



# *In situ* continuous visible and near-infrared spectroscopy of an alpine snowpack

Marie Dumont[1], Laurent Arnaud[2], Ghislain Picard[2], Quentin Libois[2,4], Yves Lejeune[1], Pierre Nabat[3], Didier Voisin[2], and Samuel Morin[1]

[1]Météo-France – CNRS, CNRM UMR 3589, Centre d'Études de la Neige, Grenoble, France
[2]UGA / CNRS, Laboratoire de Glaciologie et Géophysique de l'Environnement (LGGE) UMR 5183, Grenoble, 38041, France
[3]Météo-France – CNRS, CNRM UMR 3589, GMGEC, Toulouse, France
[4]now at: Department of Earth and Atmospheric Sciences, Université du Québec à Montréal (UQAM), Montréal, Canada

*Correspondence to:* Marie Dumont (marie.dumont@meteo.fr)

**Abstract.**

Snow spectral albedo in the visible/near-infrared range has been continuously measured during a winter season at Col de Porte alpine site (French Alps, 45.30°N, 5.77°E, 1325 m a.s.l.). The evolution of such alpine snowpack is complex due to intensive precipitation, rapid melt events and Saharan dust deposition outbreaks. This study highlights that the resulting intri­
cated variations of spectral albedo can be successfully explained by variations of the following snow surface variables : snow specific surface area (SSA), effective light-absorbing impurities content, presence of liquid water and slope. The methodology developed in this study disantangles the effect of these parameters on snow spectral albedo. The presence of liquid water at the snow surface results in a spectral shift of the albedo from which melt events can be identified with an occurence of false detection rate lower than 3.5%. Snow SSA mostly impacts spectral albedo in the near-infrared range. Impurity deposition
mostly impacts the albedo in the visible range but this impact is very dependent on snow SSA and surface slope. Our work thus demonstrates that the SSA estimation from spectral albedo is affected by large uncertainties for a tilted snow surface and medium to high impurity contents and that the estimation of impurity content is also affected by large uncertainties, especially for low values below 50 ng g$^{-1}$ black carbon equivalent. The proposed methodology opens routes for retrieval of SSA, im­
purity content, melt events and surface slope from spectral albedo. However, an exhaustive accuracy assessment of the snow
parameters retrieval would require more independent *in situ* measurements and is beyond the scope of the present study. Such time series of snow spectral albedo nevertheless already provides a new insight into our understanding of the evolution of snow surface properties.

## 1 Introduction

Snow is among the most reflective materials on Earth (Dozier et al., 2009) and its albedo exhibits large spectral variations
in the solar spectrum (Warren, 1982). Snow albedo is therefore a crucial variable of the Earth energy balance (Imbrie and Imbrie, 1980) through the surface energy budget of snow covered surfaces. Snow spectral albedo varies as a function of many



factors such as (i) the spectral and angular characteristics of the solar incident radiation and (ii) the physical and chemical properties of the snowpack (Warren, 1982; Gardner and Sharp, 2010). Since the absorption of solar energy affects in turn the physical and chemical properties of the snowpack, snow albedo is involved in several feedback loops (e.g. Flanner and Zender, 2006, Doherty et al., 2010) which generally enhance snow metamorphism and melt and are thus of crucial importance for the evolution of snow covered area and more generally for the Earth climate. Measuring, understanding and modelling snow spectral albedo is therefore essential.

The variations of snow spectral albedo with snow microstructure can be well explained using the snow surface specific area (SSA), i.e. the ratio of the surface of the ice-air interface to the mass of ice (e.g. Grenfell and Warren, 1999, Carmagnola et al., 2013). Snow spectral albedo decreases with SSA in the near-infrared so that an increased amount of solar energy is absorbed, that further accelerates snow metamorphism (Picard et al., 2012). Ice crystal habit influences the spectral albedo but this effect is not fully understood yet (e.g. Picard et al., 2009, Libois et al., 2013, Kokhanovsky and Zege, 2004, Malinka, 2014).

Light absorbing impurities (LAI) such as mineral dust, soot or algae cause a decrease in snow albedo in the visible wavelengths (Warren, 1982). This impact is enhanced at low SSA, which results in complex interdependencies and gives raise to an additional snow albedo feedback loop (Doherty et al., 2010). LAI also tend to concentrate at the surface during melt, further lowering the albedo (Sterle et al., 2013). Modelling the effect of LAI on snow spectral albedo is challenging since large uncertainties are associated with their nature, refractive indices (e.g. Dang et al., 2015) and physical properties. The size distributions and exact location with respect to the ice matrix of LAI in snow also induce large uncertainties in the simulated spectral albedo (Flanner et al., 2012).

Because the constrast between ice and water refractive indices is small, the effect of the presence of liquid water on snow spectral albedo is subtle. Warren (1982) explained that it simply increases the effective grain size. As detailed in Green et al. (2006), Dozier et al. (2009) and Gallet et al. (2014), the liquid water absorption features are shifted towards shorter wavelengths. Accurately predicting the amplitude of the shift for a given liquid water content (LWC) is somewhat difficult since it would require to know the exact location of the liquid water with respect to the ice matrix (e.g. Gallet et al., 2014). Although the retrieval of LWC from spectral albedo seems challenging, the presence of liquid water in snow can be detected using hyperspectral measurement by searching for absorption features minimum around 1000 nm(Green et al., 2006).

Slope and roughness of the snow surface also largely affect snow spectral albedo. Increased roughness generally leads to a decrease in albedo (e.g. Zhuravleva and Kokhanovsky, 2011). Slope modifies the effective incident zenith angle of solar radiation. The effect on the albedo thus depends on the ratio of diffuse to total incoming radiation (e.g. Wang et al., 2016, Dumont et al., 2011). This implies that the surface slope effect on measured snow spectral albedo varies with the wavelength along with the spectral diffuse to total solar irradiance ratio.

Snow spectral albedo is therefore extremely informative on the state of the snow "surface". Such spectral observations either from *in situ* or airborne sensors are however sparse in time and space. Dozier et al. (2009) provided an overview of the capability offered by hyperspectral remote sensing to study the evolution of optical grain size, surface liquid water and light absorbing impurities (LAI) radiative forcing. Painter et al. (2013) and Seidel et al. (2016) developed algorithms to retrieve optical grain size (which can be related to SSA *via* $r_{opt} \simeq \frac{3}{\rho_{ice}SSA}$, e.g. Gallet et al., 2009), LAI radiative forcing and broadband albedo from





airborne hyperspectral images. These data offered a new insight into spatial and temporal evolution of snow surface parameters over large areas. Carmagnola et al. (2013) and Negi and Kokhanovsky (2011) used *in situ* measured spectral albedo and related its variations to specific surface area (SSA) and LAI content. These studies all focused on spectral measurements sparse in time. More recently, Libois et al. (2015) and Picard et al. (2016a) presented a 3-year time series of spectral albedo acquired at

5 Dome C, Antarctica using an automatic spectral albedometer called Autosolexs. They developed a methodology to correct for the instrument artefacts, assess the measurements uncertainties and retrieve near-surface snow SSA.

Snow surface albedo variations at Dome C are essentially due to SSA variations because the LAI content is too low to significantly affect the albedo (Warren et al., 2006) and the snowpack remains dry throughout the year. On the contrary, in alpine conditions the presence of LAI and liquid water can significantly affect snow albedo (e.g. Di Mauro et al., 2015). This

motivates the deployment of a second Autosolexs at the Col de Porte site, French Alps to monitor the evolution of snow spectral albedo during one snow season.

The main goal of the paper is to investigate how alpine snow spectral albedo variations can be attributed to variations of surface and near-surface snow properties, namely SSA, effective impurity content, presence of liquid water and surface slope. We first investigate how an analytical formulation of spectral albedo as a function SSA, effective impurity content and slope

can be used to simulate the measured albedo. We then investigate the theoretical uncertainties associated to the optimal snow characteristics. Finally, the consistency of the time-series of optimal SSA, slope, effective impurity content and presence of liquid water with respect to independant snow and meteorological measurements is investigated. Section 2 provides an overview of the study site and instruments. Section 3 describes the method used to analyze albedo variations. Section 4 provides results and discussion.

## 20 2 Data

The measurements were taken at Col de Porte, which is located at 1325 m altitude in the Chartreuse mountain range, France. An exhaustive description of the study site and the long-term measurement instruments is provided in Morin et al. (2012). The long-term measurements used in this study are 2-m air temperature, surface temperature, direct and diffuse broadband incident shortwave radiations, snow depth and snow water equivalent, snow and rain fall rates.

### 25 2.1 Spectral albedo measurements

From January to May 2014, the Autosolexs instrument was installed at Col de Porte site (Fig. 1) to automatically measure spectral albedo. This instrument is similar to the one described in Libois et al. (2015) and Picard et al. (2016a), except that it only features one albedo head. The device acquired an upward and downward spectrum every 12 minutes over the 350-1050 nm range with an effective spectral resolution of 3 nm. Two automatic cameras provided qualitative information on the weather,

30 snow and device state during the entire season.

Contrary to Dome C site, Col de Porte site features large snow precipitation events which cover the upward looking channel. Consequently, the device was cleaned up manually after each snowfall. The snow surface below the measurements head was





not flat due to local topography. Figure 1 indeed shows that there is a slight North facing slope below the measurement head. Note that the tilt of the sensor was recorded during the measurements and remained smaller than $0.5°$.

Following Picard et al. (2016a), every acquired spectrum was corrected for (i) dark current and stray light, (ii) integration time scaling, (iii) calibration and (iv) collector angular response. The corrected spectra are used to compute the bi-hemispherical reflectance (Schaepman-Strub et al., 2006), simply called spectral albedo in the following.

## 2.2 Additional *in situ* measurements

During this snow season, additional *in situ* measurements were carried out :

– Surface SSA measurements : The surface SSA was measured on 2014-01-22 using the ASSSAP instrument (Arnaud et al., 2011) between 12:00 and 13:00 a few meters away from Autosolexs. The experimental protocol for surface SSA measurements is described in Libois et al. (2015) and Carmagnola et al. (2014).

– Impurity content measurements : Vertical profile of impurity content was measured on 2014-02-11. Refractory BC was measured using a SP2 instrument, following the procedure described in Lim et al. (2014). Insoluble dust measurements were performed using a microparticle counter (Coulter counter©Multisizer III) for particles with a diameter ranging from 1 to 30 $\mu$m, divided in 300 equivalent size channels. The total mass of dust was calculated from the volume size distribution, assuming a density of 2.5 g cm$^{-3}$ (Delmonte et al., 2002).

– LWC measurements : Vertical LWC profile was measured for the whole snowpack on a weekly basis at Col de Porte site using a dielectric probe at 13 MHz (Brun et al., 1989). Fifteen measurements were available within the period of observation of the albedo (4 in January, 4 in February, 4 in March and 3 in April).

## 2.3 Atmospheric model outputs

Knowlegde of the spectral diffuse to total solar irradiance ratio and of atmospheric aerosol deposition fluxes are crucial to understand the evolution of the measured snow spectral albedo. In this respect, we used ouputs from the atmospheric model ALADIN-Climate at Col de Porte site to calculate diffuse and direct spectral solar irradiance and to investigate the temporal evolution of dry and wet aerosols deposition fluxes.

ALADIN-Climate is a regional climate model based on a bi-spectral semi-implicit semi-Lagrangian scheme. The version 5.3 (Nabat et al., 2015) is used in the present study with a 50 km horizontal resolution, 31 vertical levels and the ERA-Interim reanalysis (Dee et al., 2011) as lateral boundary forcing. This model includes a prognostic aerosol scheme, adapted from the GEMS/MACC aerosol scheme (Morcrette et al., 2009 ; Benedetti et al., 2011 ; Michou et al., 2015). The main aerosol species are represented: dust, sea-salt, sulphate, black carbon and organic particles. Natural aerosols (dust and sea-salt) are emitted from the surface depending on surface wind and soil characteristics, while anthropogenic emissions come from external inventories (Lamarque et al., 2010). The spatial domain of our simulations has been designed to include all the sources generating aerosols that can be transported over the French Alps, such as the Saharan desert and a large part of the Northern Atlantic Ocean. However, aerosols coming from longer-range transport (e.g. fires in America) are not considered.



More details about the aerosol scheme, as well as an evaluation showing the performance of the scheme, can be found in Nabat et al. (2015). All these aerosols interact with the shortwave and longwave radiation scheme. Even if ALADIN-Climate is a regional climate model, the model has the ability to reproduce the observed weather chronology thanks to the spectral nudging method (Radu et al., 2008), which enables us to keep large scales atmospheric conditions from the boundary forcing

5 and thus impose the observed unfolding of the meteorological conditions that is essential to represent dust events. In this simulation, surface pressure, wind vorticity and divergence and specific humidity were nudged towards ERA-Interim.

The outputs of the ALADIN-Climate model used in this study are hourly total aerosols optical thickness, total water vapour column and total ozone column and dust and black carbon wet and dry deposition fluxes.

## 3   Methods

### 3.1   Estimation of the direct to diffuse solar irradiance ratio

The ratio of diffuse over direct irradiance is required to perform accurate correction of the measured spectrum (e.g. Picard et al., 2016a). For this purpose and to overcome the coarse resolution of ALADIN-Climate we use hourly 2 m air temperature and the broadband direct ($SW_{dir}$) to total ($SW_{tot}$) ratio directly estimated from Col de Porte measurements (Morin et al., 2012). The ratio is used to estimate hourly cloud optical thickness at 550 nm, $\tau$.

For this purpose, the SBDART detailed radiative model (Ricchiazzi et al., 1998) was used to first calculate $SW_{dir}$ over $SW_{tot}$ from varying cloud optical thicknesses and mean atmospheric conditions at Col de Porte (aerosols optical thickness, total ozone column and total water vapour column). A regression equation (Eq. 1) was derived from those results and used to estimate $\tau$ from $SW_{dir}$ over $SW_{tot}$ ratio measured at Col de Porte.

$$\tau = \mu_s \log\Big(\frac{1 + 0.8\mu_s - 1.95\mu_s^2 + 1.67\mu_s^3 - 0.77}{1 - SW_{dir}/SW_{tot}}\Big), \qquad (1)$$

where $\mu_s = \cos\theta_s$ is the cosine of the solar zenith angle.

The outputs of the ALADIN-Climate (namely hourly total aerosols optical thickness, total water vapour column and total ozone column) together with measured air temperature and estimated $\tau$ were used as inputs for the SBDART (Ricchiazzi et al., 1998) model in order to compute the hourly diffuse and direct spectral irradiance. The ratio between these variables is used in our analysis.

### 3.2   Spectral albedo dependence on snow SSA and impurity content

In order to relate albedo to snow properties, we use the theoretical formalism of Kokhanovsky and Zege (2004). Several assumptions are made. (i) The snowpack is horizontally and vertically homogeneous, which means only one bulk SSA and impurity content value are used to explain albedo variations. (ii) The surface is flat. (iii) Snow phase function and single scattering albedo are implicitly described by the asymmetry factor, $g$, the absorption enhancement parameter, $B$, and SSA. (iv)

The surface and the sensor are perfectly horizontal. The effect of surface slope is investigated in Section 3.3.





Under these conditions, following Libois et al. (2013) and Picard et al. (2016a), snow albedo can be written as

$$\alpha_{\text{th}}(\lambda,\theta) = r_{\text{diff}}(\lambda,\theta)\alpha_{\text{diff}}(\lambda) + (1 - r_{\text{diff}}(\lambda,\theta))\alpha_{\text{dir}}(\lambda,\theta) \tag{2}$$

$$\alpha_{\text{diff}}(\lambda) = \exp\Big(-\sigma(\lambda,\text{SSA},c_{\text{imp}})\Big) \tag{3}$$

$$\alpha_{\text{dir}}(\lambda,\theta) = \exp\Big(-\frac{3}{7}(1+2\cos\theta)\sigma(\lambda,\text{SSA},c_{\text{imp}})\Big) \tag{4}$$

$$\sigma(\lambda,\text{SSA},c_{\text{imp}}) = \sqrt{\frac{64\pi}{3\lambda\rho_{\text{ice}}\text{SSA}(1-g)}\left(2n_i(\lambda)B + \frac{3\rho_{\text{ice}}c_{\text{imp}}}{\rho_{\text{imp}}}\mathcal{I}m\left(\frac{m_{\text{imp}}^2 - 1}{m_{\text{imp}}^2 + 2}\right)\right)} \tag{5}$$

where $r_{\text{diff}}(\lambda,\theta)$ is the diffuse to total irradiance ratio, $\rho = 917\,\text{kg}\,\text{m}^{-3}$ is the ice density at 0 °C and $n_i(\lambda)$ is the imaginary part of the ice refractive index taken either from Warren and Brandt (2008) (default) or from Picard et al. (2016b). $B=1.6$ and $g=0.85$ are constant and taken from Libois et al. (2014). $m_{\text{imp}}$ is the complex refractive index of BC taken from Flanner et al. (2012). $\rho_{\text{imp}}$ is BC density and is set to $1270\,\text{kg}\,\text{m}^{-3}$ according to Flanner et al. (2012). $c_{\text{imp}}$ is the effective impurity mass per unit of snow mass ($\text{kg}\,\text{kg}^{-1}$) where effective stands for BC optically equivalent content.

Following Picard et al. (2016a), we introduce a scaling factor A to account for several artefacts and shortcomings in the measurement technique. This scaling factor relates the measured albedo, $\alpha_{\text{meas}}$ and the theoretical albedo obtained from Eq. 2, $\alpha_{\text{th}}$.

$$\alpha_{\text{meas}}(\lambda,\theta) = A\alpha_{\text{th}}(\lambda,\theta) \tag{6}$$

## 3.3 Effect of the slope on measured albedo

The slope of the snow surface introduces a change in the solar irradiance with respect to the solar radiation incoming on a perfectly horizontal surface. This change is of crucial importance for our application since it is wavelength-dependent. If we assume that (i) both diffuse solar radiation and reflected radiation are isotropic and (ii) the surface slope is small and local enough not to modify significantly the solid angles under which the incoming and reflected radiations are measured with respect to what would happen for an horizontal surface, then the slope of the surface only affects the effective sun zenith and azimuth angles and thus the direct solar irradiance (see details in App. A and B). Note that for fully cloudy days, under these assumptions, the slope has consequently no influence on the measured albedo.



Let $\theta_s$ and $\phi_s$ be the direction of sun with respect to a perfectly horizontal surface, $\theta_n$ and $\phi_n$ being respectively the slope and aspect of the surface and $\tilde{\theta}_s$ and $\tilde{\phi}_s$ the effective direction of the sun with respect to the tilted surface. Then $\cos\tilde{\theta}_s = K\cos\theta_s$ (e.g. Dumont et al., 2011) where

$$K = \cos\theta_n + \tan\theta_s \sin\theta_n \cos(\phi_s - \phi_n) \tag{7}$$

This leads to the following relationship between the measured and the theoretical albedo on an horizontal surface (see details in Appendices A and B) :

$$\alpha_{\mathrm{meas}}(\lambda,\theta_s) = A\left( r_{\mathrm{diff}}(\lambda,\theta_s)\alpha_{\mathrm{diff}}(\lambda) + (1 - r_{\mathrm{diff}}(\lambda,\theta_s))K\alpha_{\mathrm{dir}}(\lambda,\tilde{\theta}_s) \right) \tag{8}$$

The variations of measured spectral albedo with SSA, $c_{\mathrm{imp}}$ and surface slope are illustrated in Figure 2a. The effect of the anisotropy of diffuse solar radiation is discussed in Bogren et al. (2016) while the effet of the anisotropy of the reflected radiation is discussed in Dumont et al. (2010) and Carmagnola et al. (2013). The anisotropy of the sky diffuse component and of the reflected radiation is second order as long as the cosine response correction is small (Picard et al., 2016a, Carmagnola et al., 2013).

### 3.3.1 Albedo variations analysis method

In order to relate variations of spectral albedo to variations of surface snow properties, we apply the following methodology to the measured albedo.

*Step 1 : Estimate the scaling factor A*

A seasonal value of A is estimated using Eq. 8 with $r_{\mathrm{diff}} = 1$ (selected fully cloudy days from visual inspection of the camera photographies and $\theta_s$ smaller than 65°) and a nonlinear least squares method (provided by Python scipy.optimize.leastsq function). To avoid undetermination problem between A and $c_{\mathrm{imp}}$ in case of moderate to high amount of impurities in snow, only the spectra from the beginning of season were used for this estimation.

Quantiles 25 and 75 of A distribution are used to propagate uncertainties on the near surface and surface properties predicted from the spectra.

*Step 2 : Estimate optimal SSA and $c_{\mathrm{imp}}$*

Once the value of A is set, optimal SSA and $\log_{10}(c_{\mathrm{imp}})$ values are estimated from the spectrum within 400-1050 nm along with $K$ using Eq. 8 and the non linear least square optimization method. For cloudy days, the optimization is performed with $K=1$. Clear sky days are selected as days with $\tau<0.01$, $\tau$ being estimated using Eq. 1.

After these steps, spectra are filtered based on the root mean square deviation (RMSD) between calculated and measured spectrum with a threshold of 0.022. This value was set to account for discrepancies in the visible wavelengths due to presence of red dust which is common at Col de Porte. Illustration of *Step 2* is provided for three spectra in Fig. 2b. The black spectrum shows a clear "dusty" pattern in the visible wavelengths (400-500 nm).





Note that by using a seasonal A value for every spectrum, we assume that the measurement artifacts are the same under cloudy and clear sky conditions except for the effect of slope.

*Step 3 : Estimate daily optimal surface slope and aspect*

Using $K$ diurnal cycles and Equation 7, we estimate daily optimal values of surface slope angle and aspect for fully clear sky days. This is not stricly needed but it indirectely validates that $K$ optimization has not compensated for other artifacts than slope.

*Step 4 : Detect if the snow surface is wet or dry*

Taking benefit of the spectral shift of the absorption feature at 1030 nm in presence of liquid water (Green et al., 2006), we apply the following method to distinguish wet from dry snow. The spectra are first filtered using a 20 nm moving window average in order to reduce noise before minimum calculation. We then compute the wavelength of minimum albedo in the 1000-1050 nm range and apply the following criteria to detect liquid water presence :

$$\underset{\lambda \in [1000,1050]\,\mathrm{nm}}{\mathrm{argmin}} (\alpha(\lambda)) < \lambda_{\mathrm{water}} \tag{9}$$

$\lambda_{\mathrm{water}}$ threshold has been set after studying the distribution of $\mathrm{argmin}(\alpha)$ on the whole spectra dataset.

## 4   Results and discussion

### 4.1   Determination of the experiment specific parameters A, $\theta_n$, $\phi_n$ and $\lambda_{\mathbf{water}}$

#### 4.1.1   Determination of scaling factor A

Figure 3 describes the distribution of A for fully cloudy days before 2014-03-05. The distribution has a median value 0.943, only slightly lower than the ideal value of one. The spread of A values is small (quantile 25, $q_{25}$ : 0.920 and quantile 75, $q_{75}$ : 0.964). This spread could probably be attributed to residual measurement artifacts such as deposition of precipitation particles on the measurement head, presence of small direct incoming radiation, and reproductibility errors due to the optical switch (Picard et al., 2016a).

In the following, A is set to 0.943. Quantiles 25 and 75 are used to propagate uncertainties of the near surface and surface properties predicted from the spectra, i.e. the optimal parameters.

#### 4.1.2   Determination of slope angle and aspect

Figure 4a illustrates that the optimal parameter $K$ (green diamonds) follows a diurnal cycle with higher values in the morning and lower values in the afternoon as predicted by Equation 7. It also illustrates the good agreement between the optimal $K$ and simulated $K$ (blue crosses, *step 3*).

Figure 4b shows the daily estimates of $\theta_n$ and $\phi_n$ obtained after *step 3* over the season. The estimated slope varies from 3 to 10 ° while the aspect mainly ranges between 300 and 360 ° (discarding obvious outlier values). The uncertainty range of estimated $\theta_n$ is generally lower than 2° while larger uncertainty ranges, up to 50°, are estimated for aspect. The average slope





and aspect are in agreement with what can be visually estimated from Figure 1. The seasonal evolution of slope and aspect seems to be related to snow evolution (the red line in Figure 4b shows the measured snow depth). Slope is smaller just after a precipitation event and increases during the melt season.

For comparison, the same method has been applied to the albedo database measured in Dome C and described in Picard et al. (2016a). The snow surface is horizontal at large scale but the surface can be locally rough due to wind-drift effects. The method applied to Dome C data during season 2012-2013 leads to slope angle between $\pm\,2°$ thus providing an insight into the accuracy of the method.

### 4.1.3 Determination of $\lambda_{\text{water}}$

Figure 5b shows the distribution of the wavelength at which the minimum reflectance value is reached for all the measured spectra with RMSD lower than 0.022 over the 400-1050 nm range. The distribution is bimodal as predicted by Green et al. (2006) and exhibits two peaks, the first one at 1029 nm and the second one at 1034 nm. The first one is likely to correspond to wet snow, while the second is for dry snow. $\lambda_{\text{water}}$ in Eq. 9 is consequently set to 1032 nm in the following. Figure 5a provides an illustration for two spectra (a wet and a dry one) measured on 2014-03-09.

### 4.2 Theoretical uncertainty and representativeness of the estimated snow parameters

Section 3.3.1 describes the methodology applied in this study to estimate optimal SSA, $c_{\text{imp}}$ and presence of liquid water from the measured spectra. The uncertainty related to the optimal SSA is discussed in detailed in Picard et al. (2016a) together with the vertical representativeness of the estimated SSA. However our study differs from Picard et al. (2016a) since (i) the snowpack contains larger amount of light absorbing impurities, (ii) the snow surface is not perfectly horizontal and (iii) the snow can be wet. These points might indirectly affect the estimated uncertainty of the optimal SSA and of the other snow parameters, which is investigated in the section below.

Note that the results presented below in Sections 4.2.1 and 4.2.2 have been done under clear sky conditions. The methodology applied in this study for cloudy sky intrinsically leads to larger uncertainties for cloudy sky conditions.

### 4.2.1 Effect of slope on the optimal parameters

Figure 6a shows an example of measured spectrum on 2014-03-10 (grey solid lines) compared to the optimal spectrum predicted with slope (i.e. from *Step 2*, red dotted lines), assuming a perfectly horizontal surface (blue dotted lines) and slightly varying slope zenith (resp. azimuth) angles from $+2°$ (resp. $+10°$) (green dotted line). It illustrates that the (i) the shape of the spectrum computed with no slope does not agree with the measured albedo, (ii) the best agreement is obtained for the red dotted line (i.e. with slope) and (iii) a small variation of the slope angles induces a variation of 10% for the optimal SSA and of 13 ng g$^{-1}$ for optimal $c_{\text{imp}}$.

More generally, Fig. 4b shows that the spread of A distribution is translated in an uncertainty of typically less than 2° for slope zenith angle and of less than 10° for the aspect. Using the simulated spectrum from Fig. 2 and adding slope and aspect





variations within these ranges, we obtain retrieved SSA variations up to 20% for 40 m$^2$ kg$^{-1}$ and up to 10% for 5 m$^2$ kg$^{-1}$. The larger uncertainty associated with the higher SSA value is because tilt effect is proportional to the albedo value (higher for higher SSA). Optimal $c_{imp}$ variations are up to 25 (resp. 40) ng g$^{-1}$ for SSA=40 m$^2$ kg$^{-1}$, $c_{imp}$=0 (resp. 100) ng g$^{-1}$. For lower SSA (5 m$^2$ kg$^{-1}$), $c_{imp}$ variations range from 4 to 40 ng g$^{-1}$ for initial impurity content ranging from 0 to 1000 ng g$^{-1}$.

In addition, the surface slope also affects the value of $\mathrm{argmin}(\alpha)$. Using the spectral albedo from Fig. 2 and slope angle varying between -20 and +20°, we nevertheless found that $\mathrm{argmin}(\alpha)$ does not vary for slope angle lower than 10° and that the maximum variation for steeper slope is less than $\pm 0.5$ nm.

### 4.2.2    Coupled effect of SSA and $c_{imp}$ on spectral albedo

Figure 6b shows an example of measured spectrum on 2014-04-03 (grey solid lines) compared to the optimal spectrum pre-
dicted with slope (i.e. from *Step 2*, red dotted lines). The blue dotted line corresponds to the optimal albedo obtained while increasing by 50 ng g$^{-1}$ the original optimal value of $c_{imp}$ and the green dotted line to the optimal albedo obtained while adding +15 % to the original optimal value of SSA. It illustrates that changes in the SSA (resp. $c_{imp}$) induce changes in the optimal values of $c_{imp}$ (resp. SSA).

More generally, using again the simulated spectrum presented in Fig. 2, we first investigate the change in estimated SSA
while varying $c_{imp}$ by $\pm$ 50 ng g$^{-1}$. The larger variations, up to 15%, are obtained for very low $c_{imp}$ (1 ng g$^{-1}$) and small SSA (5 m$^2$ kg$^{-1}$). As soon as $c_{imp}$ is higher than 100 ng g$^{-1}$ the SSA variations are smaller than 13 % for SSA=5 m$^2$ kg$^{-1}$ and than 5 % for higher SSA (40 m$^2$ kg$^{-1}$). Then, we investigate the change in optimal $c_{imp}$ while varying SSA by $\pm$ 15 %. It shows that a $\pm$ 15 % uncertainty on SSA leads to a $\pm$ 20 % uncertainty on $c_{imp}$.

### 4.2.3    Optimal SSA and liquid water

The presence of liquid water modifies the spectrum. Thus it necessarily affects the value of optimal SSA estimated from the spectrum. Gallet et al. (2014) provides an overview on how it affects the SSA estimated from 1310 nm reflectance. The conclusions is different in our case where the whole spectrum is used. Although a detailed modelling study is beyond the scope of the study, using the relative difference in optimal SSA, $\Delta\mathrm{SSA} = \frac{\mathrm{SSA_{wet}} - \mathrm{SSA_{dry}}}{\mathrm{SSA_{wet}}}$, between two consecutive spectra (absolute time difference of 12 min), we were able to investigate this effect. The distribution of $\Delta\mathrm{SSA}$ for the whole measurements
period is peaked. Quantiles 25 (resp. 50 and 75) are -4.15 % (resp. 0.025 % and 6.99 %). Assuming the change in SSA between the two spectra is only due to refreezing and not to metamorphism, this indicates that the SSA of the wet spectrum is not generally higher or smaller than the SSA of the dry spectrum and that an upperbound of the uncertainty on the optimal SSA due to the presence of liquid water is 7%.




### 4.2.4 Vertical representativeness of optimal impurity content

Real snowpacks are not vertically homogeneous as assumed in our method and often displays high vertical gradient of SSA or impurity content (e.g. Sterle et al., 2013). However only one bulk SSA and $c_{imp}$ values are estimated is order to stabilize the optimization.

The vertical representativeness of SSA is discussed in Picard et al. (2016a). Using the same methodology, we analyze here the vertical representativeness of $c_{imp}$. We use the two-stream radiative model TARTES (Libois et al., 2013) to compute the albedo of semi-infinite medium with a fixed SSA, density and $c_{imp}$. A layer with variable snow water equivalent $h$ is added on top of it with a slightly different value of $c_{imp}$ ($c_{imp} + \delta c$). The relative contribution of the uppermost layer to the albedo is then defined as $(\alpha(h) - \alpha(0))/(\alpha(\infty) - \alpha(0))$ where $\alpha(h)$ is the albedo computed with TARTES averaged over 400-1050 nm for a two layers snowpack with the uppermost layer of swe h. The value of $\delta c$ does not change the results as long as it is small (e.g. + 10% in Fig. 7). Thus, the vertical representativeness calculated here is only valid for slightly inhomogenous snowpack.

Figure 7 shows the relative contribution of the uppermost layer as a function of its SWE for two SSA values (5 m$^2$ kg$^{-1}$ plain lines, 40 m$^2$ kg$^{-1}$ crosses) and varying values of $c_{imp}$ (from 1 to 500 ng g$^{-1}$). It shows that (i) the higher the SSA and $c_{imp}$, the higher the contribution of the uppermost centimeters. For low SSA (5 m$^2$ kg$^{-1}$), high density (400 kg m$^{-3}$) and $c_{imp}$ (500 ng g$^{-1}$), the uppermost 5 cm contributes to more than 80 % of the signal. For higher SSA (40 m$^2$ kg$^{-1}$), lower $c_{imp}$ (10 ng g$^{-1}$) and density (200 kg m$^{-3}$), the uppermost 15 cm contributes to more than 80 % of the signal.

## 4.3 Time evolution of the optimal SSA

Sections 4.3 and 4.4 focus on of the optimal SSA and effective impurity content time series predicted from the measured spectra and their consistency with snow and meteorological conditions.

### 4.3.1 Seasonal evolution of "noon" SSA

Figure 8 a,b presents the seasonal evolution of all the optimal SSA and the optical grain size, $r_{opt}$ predicted from the measured spectra between 12:00 and 13:00 each day. Clear sky conditions are represented in blue and cloudy conditions in red. Vertical bars represent the uncertainties derived from $q_{25}$ and $q_{75}$ of A distribution. The SSA ranges from more than 70 m$^2$ kg$^{-1}$ after snowfall down to 2 m$^2$ kg$^{-1}$. For clear sky days uncertainties are small (typically less than 10 %) whereas cloudy days feature larger uncertainties. The range of variations is consistent with measured values of surface SSA presented in Morin et al. (2013) using the DUFISSS instrument (Gallet et al., 2009) in 2010. The SSA decay is also consistent with current understanding and parameterizations of SSA evolution (e.g. Carmagnola et al., 2014, Flanner and Zender, 2006, Schleef et al., 2014).

On 2014-01-22, surface SSA was measured between 12:00 and 13:00 with ASSSAP at $44 \pm 4$ m$^2$ kg$^{-1}$. The sky was partially clear during the day and the sensor was covered by snow in the morning due to recent precipitation leading to valid spectra only after 13:00. The optimal SSA value at 13:00 is (42.6, 43.1, 43.4) m$^2$ kg$^{-1}$ ($q_{25}$, $q_{50}$, $q_{75}$) which is only slightly lower than ASSSAP value and within ASSSAP uncertainty range.





Note that the small SSA increase (mostly visible in Fig. 8b) by 2014-04-10 is an artifact induced by the appearance of grass in the field of view of the sensor, leading to higher reflectance in NIR wavelengths and thus higher optimal SSA.

### 4.3.2 SSA diurnal cycles

Figure 9a,b shows the diurnal evolution of SSA inferred from albedo variations. Figure 9a zooms on the details of the diurnal cycle from 2014-03-06 to 2014-03-09. The diurnal cycle evolves from higher SSA in the morning to lower SSA in the afternoon in the absence of precipitation which can be explained by snow metamorphism and by the snow intrinsic albedo feedback (Dumont et al., 2014).

Additionnally Fig. 9c presents the daily SSA decay rate in % per day inferred from the diurnal cycle. The grey area corresponds to the 15 % uncertainty estimated in Picard et al. (2016a). SSA decay rates are larger after snowfall. Such variations are also described by Seidel et al. (2016) when investigating $r_{opt}$ variations from airborne hyperspectral imagery.

### 4.4 Time evolution of the optimal effective impurity content

Figure 8b describes the evolution of the effective impurity content, $c_{imp}$ inferred from spectral albedo values. The values are affected by large uncertainties for cloudy days as already discussed for SSA. $c_{imp}$ values range between 0 and 1000 ng g$^{-1}$. Values up to 400 ng g$^{-1}$ are found in January and February. The highest values are found at the end of the season after long periods without precipitation. $c_{imp}$ variations range is in agreement with other studies conducted in the Alps (e.g. Di Mauro et al., 2015). Low impurity content (typically smaller than 50 ng g$^{-1}$) are affected by large uncertainties.

On Februray 11th, the equivalent BC content measured in the first two cm of the snowpack was $17 \pm 6$ ng g$^{-1}$. The day was partially cloudy. Only two spectra before 12:00 are valid and measured during clear sky conditions (from the camera images). For these spectra, the optimal SSA values are [22, 23, 24] m$^2$ kg$^{-1}$ and $c_{imp}$=[0.5, 2., 6.6] ng g$^{-1}$ ($q_{25}$, $q_{50}$, $q_{75}$). The measured $c_{imp}$ is higher than the optimal one though lying in the range of impurity content values for which the retrieval is highly uncertain.

Diurnal cycle of $c_{imp}$ is not investigated in this study since the uncertainties associated with the $c_{imp}$ values are too high with respect to the diurnal variations.

Figure 10b shows the seasonal evolution of $c_{imp}$ but distinguishes the prevailing color of impurities (black : black crosses or red: red dots). This distinction is based on the RMSD calculated between the modeled and the measured albedo within the 400-500 nm wavelengths range. If this RMSD is larger than the RMSD over the whole spectrum, $c_{imp}$ is represented in red (see for example the 2014-04-04 albedo in Fig. 2b). The blue vertical bar represents days for which melt was detected on the SWE measurements (Fig. 10a). Figure 10c shows the wet and dry deposition predicted by ALADIN-Climate for black carbon and mineral dust at Col de Porte site.

High values of $c_{imp}$ in January and February may be related to dust deposition events. Low values are found beginning of March after a precipitation period. From March to mid-April, the exponential increase in $c_{imp}$ can be related to melt during which part of the impurities concentrate on the surface (Sterle et al., 2013) and to dust deposition events beginning of April. Note that the coarse resolution of ALADIN-Climate limits the accuracy of the precipitation events. Consequently the wet



deposition event predicted on 2014-03-30 has no impact on the snowpack because no precipitation were observed during that day at Col de Porte.

Finally, the uncertainty associated with the imaginary part of the refractive index of ice (e.g. Warren and Brandt, 2008, Carmagnola et al., 2014, Picard et al., 2016b) strongly affects the relationship between spectral albedo and $c_{imp}$ (see Eq. 5). Figure 10b shows the values of $c_{imp}$ estimated using the values of the ice refractive index proposed in Picard et al. (2016b) instead of Warren and Brandt (2008) (green dots). $c_{imp}$ are significantly lower using Picard et al. (2016b) especially for low impurity content.

## 4.5 Presence of liquid water

Figure 11a shows the comparison between the wavelength of minimum reflectance and the surface LWC measured in the field for 15 snow pits. The agreement between our method of detection and field observations is perfect for these 15 dates.

Figure 11b, c shows the results of the liquid water detection method over the whole season. The detection is in good agreement with measured surface temperature. The accuracy of the measured surface temperature is estimated equal to 1 K. False-detection cases, i.e. when the snow is detected as wet but surface temperature is negative, i.e. less than 272.15 K to account for the measurement accuracy, are less than 3.5 %. False detection cases may originate from measurement artifacts since the signal to noise ratio for the considered wavelengths is high and differences in the field of view of the spectrometer and of the longwave downlooking sensor used to calculate surface temperature. Figure 11c shows that melt-freeze diurnal cycles are well depicted with dry snow in the morning and late afternoon and wet snow around noon.

## 5 Concluding remarks

This study introduces a continuous data set of spectral albedo acquired from January to May 2014 from 400 to 1050 nm at the Col de Porte alpine site as a follow-up of the study conducted using the same instrument at Dome C in Antarctica during 3 years (Picard et al., 2016a). The alpine snowpack radically differs from inner Antarctica snowpack and the spectral albedo variations cannot be only explained by the evolution of near surface SSA and solar zenith angle. We investigated how the variations of measured spectral albedo can be explained by the evolution of other factors such as effective impurity content, surface slope and aspect and presence of liquid water. For this, we used analytical formulation of spectral albedo as a function of these near surface parameters. Results show that the measured spectra are accurately simulated using this formulation. This formulation disentangles the effect of slope, SSA and effective impurity content on spectral albedo and consequently provides optimal values of these parameters for each measured spectra. This study also demonstrates that wet and dry snow surfaces can be distinguished taking benefit of the spectral shift between the refractive index of ice and water around 1000 nm.

The uncertainty associated to the optimal SSA was discussed in detail in Picard et al. (2016a) and theoretically estimated to be better than 15%. In our case, in presence of impurities and surface slope, the accuracy is degraded. The detection of wet snow surface (only binary information) at the snow surface is less affected by measurement artefacts and surface tilt. Finally, the estimation of an effective impurity content is subject to large uncertainties making $c_{imp}$ values lower than 50 ng g$^{-1}$

challenging to estimate as pointed out by Warren (2013). The study emphasizes the need for measuring snow albedo over snow surface as flat as possible in order to be able to infer accurately snow surface variables. The influence of surface roughness and incident and reflected radiations anisotropy deserves future work. It must be underlined that the detailed study of the accuracy of the estimation of near surface snow parameters from spectral albedo would require more extensive validation measurements

than presented in this paper, i.e. micro-tomography evaluation of SSA and systematic measurements of surface impurities content.

However, optimal near-surface SSA predicted from the spectra exhibits strong seasonal and diurnal cycles that can be related to snow and meteorological conditions. Near surface optimal effective impurity content also exhibits strong variations along the season that can be related to surface enrichment process due to melt and Saharan dust deposition events that frequently

occur in the French Alps (Di Mauro et al., 2015).

Such time series of spectral albedo are required to understand and quantify the evolution of snow albedo in relationship with snow surface variables such as SSA and surface impurity content. They are also a unique opportunity to better understand the evolution of near-surface SSA and effective impurity content during the snow season. They provide unique dataset to evaluate and refine detailed snowpack model as Crocus (Vionnet et al., 2012). Furthermore, without the need of a retrieval methodology

for snow parameters, such measured spectral albedo time-series can be assimilated in snowpack model in order to improve simulation of the snowpack structure (Charrois et al., 2016). This study hence emphasizes the usefullness of hyperspectral optical observations of snow.

## 6 Data availability

Processed spectral albedo time series will be available through Pangaea database.

*Author contributions.* M. Dumont coordinated the study and developed the retrieval algorithm. G. Picard and L. Arnaud developed and built the automatic albedometer. G. Picard, L. Arnaud, S. Morin and D. Voisin deployed it at Col de Porte. D. Voisin performed the impurity content measurements. P. Nabat provided the ALADIN-Climate simulations and Y. Lejeune provided the snow and meteorological measurements at Col de Porte. M. Dumont prepared the manuscript with contributions from the other authors.

*Acknowledgements.* CNRM/CEN and LGGE are part of Labex OSUG@2020 (investissement d'avenir - ANR10 LABX56). This study was
25 supported by the ANR program 1-JS56-005-01 MONISNOW and the LEFE programs BON and ASSURANCE. The authors are grateful to the Col de Porte staff for ensuring a proper working of all the instruments, to L. Mbemba for the impurity content *in situ* measurements and to V. Vionnet and H. Loewe for helpful discussion on sky view factor.



## Appendix A: Diffuse radiation on a slope

Let $\theta_s$ and $\phi_s$ be the direction of sun with respect to a perfectly horizontal surface, $\theta_n$ and $\phi_n$ being respectively the slope and aspect of the surface and $\tilde{\theta}_s$ and $\tilde{\phi}_s$ the effective direction of the sun with respect to the tilted surface. Then $\cos\tilde{\theta}_s = K\cos\theta_s$ (e.g. Dumont et al., 2011) where

$$K = \cos\theta_{\mathrm{n}} + \tan\theta_{\mathrm{s}}\sin\theta_{\mathrm{n}}\cos(\phi_{\mathrm{s}} - \phi_{\mathrm{n}}) \tag{A1}$$

The fraction of diffuse irradiance, $V$, which is seen by the tilted surface can be written, in the tilted surface reference frame :

$$V = \frac{1}{\pi}\int_0^\pi \int_0^{H(\phi)} \cos\theta \sin\theta \, d\theta \, d\phi \tag{A2}$$

where $H(\phi)$ is the elevation of the horizon and is equal to $\frac{\pi}{2}$ for $\phi \in [\pi, 2\pi]$ and verifies Eq. A3 for $\phi \in [0, \pi]$.

$$K = 0, \quad \tan H(\phi) = -\frac{1}{\tan\theta_n \cos(\phi)} \tag{A3}$$

This leads to :

$$V = \frac{1}{2} + \frac{1}{\pi}\int_0^{\frac{\pi}{2}} \frac{1}{\tan^2\theta_n \cos^2\phi + 1} \, d\phi \tag{A4}$$

Finally using $X = \frac{\tan\phi}{\sqrt{1+\tan^2\theta_n}}$ in the integral above leads to :

$$V = \frac{1 + \cos\theta_n}{2} \tag{A5}$$

as in e.g. Wang et al. (2016).

## Appendix B: Effect of slope on measured albedo

The solar total incoming radation on a perfectly horizontal surface, $E\downarrow(\lambda, \theta_s)$ can be written as a function of incoming direct radiation, $E_{\mathrm{direct}}(\lambda)$ and incoming diffuse radiation, $E_{\mathrm{diffuse}}(\lambda)$ :

$$E\downarrow(\lambda, \theta_s) = E_{\mathrm{direct}}(\lambda)\cos\theta_s + E_{\mathrm{diffuse}}(\lambda) \tag{B1}$$

Let $r_{\mathrm{diff}}(\lambda, \theta_s)$ be the ratio of diffuse to total irradiance, $r_{\mathrm{diff}}(\lambda, \theta_s) = \frac{E_{\mathrm{diffuse}}(\lambda)}{E(\lambda, \theta_s)}$ Under the assumption of perfectly istropic diffuse irradiance, the solar irradiance on a tilted surface $\tilde{E}(\lambda, \theta_s)$ verifies (e.g. Wang et al. (2016)) :

$$\tilde{E}\downarrow(\lambda, \theta_s) = E_{\mathrm{direct}}(\lambda)K\cos\theta_s + E_{\mathrm{diffuse}}(\lambda)V + (1-V)\alpha_{\mathrm{true}}(\lambda, \theta_s)\tilde{E}\downarrow(\lambda, \theta_s) \tag{B2}$$

where $\alpha_{\mathrm{true}}(\lambda, \theta_s)$ is the albedo of the tilted surface and can be written :

$$\alpha_{\mathrm{true}}(\lambda, \theta_s) = K(1 - r_{\mathrm{diff}}(\lambda, \theta_s))(\alpha_{\mathrm{dir}}(\lambda, \tilde{\theta}_s) - \alpha_{\mathrm{diff}}(\lambda)) + \alpha_{\mathrm{diff}}(\lambda) \tag{B3}$$





If we assume that the reflected radiation is also isotropic, the upward radiation measured by the horizontal sensor above the tilted surface is equal to :

$$E_{\mathrm{meas}}\uparrow(\lambda,\theta_s) = \alpha_{\mathrm{true}}(\lambda,\theta_s)V\tilde{E}\downarrow(\lambda,\theta_s) + (1-V)(\tilde{E}\downarrow(\lambda,\theta_s) - E_{\mathrm{direct}}(\lambda)K\cos\theta_s) \tag{B4}$$

The instrument thus measures :

$$\alpha_{\mathrm{meas}}(\lambda,\theta_s) = A\left(K(1-\mathrm{r}_{\mathrm{diff}}(\lambda,\theta_s))\left(V\alpha_{\mathrm{dir}}(\lambda,\tilde{\theta}_s) + \frac{(1-V)\alpha_{\mathrm{true}}(\lambda,\theta_s)}{1-(1-V)\alpha_{\mathrm{true}}(\lambda,\theta_s)}(1-V+\alpha_{\mathrm{diff}}(\lambda))\right) + \frac{V\mathrm{r}_{\mathrm{diff}}(\lambda,\theta_s)}{1-(1-V)\alpha_{\mathrm{true}}(\lambda,\theta_s)}\left(V\alpha_{\mathrm{diff}}(\lambda)+1-V\right)\right) \tag{B5}$$

Let $\varepsilon^2 = 1 - V$, when $\varepsilon$ is small (e.g. $\theta_n$ is small), Eq. B5 leads to the following expression for $\alpha_{\mathrm{meas}}(\lambda,\theta_s)$:

$$\frac{\alpha_{\mathrm{meas}}(\lambda,\theta_s)}{A} = (1-\mathrm{r}_{\mathrm{diff}}(\lambda,\theta_s))\alpha_{\mathrm{dir}}(\lambda,\tilde{\theta}_s) + \mathrm{r}_{\mathrm{diff}}(\lambda,\theta_s)\alpha_{\mathrm{diff}}(\lambda)$$
$$+2\varepsilon\tan\theta_s\cos\phi(1-\mathrm{r}_{\mathrm{diff}}(\lambda,\theta_s))\alpha_{\mathrm{dir}}(\lambda,\tilde{\theta}_s)$$
$$+\varepsilon^2\left((1-\mathrm{r}_{\mathrm{diff}}(\lambda,\theta_s))(\alpha_{\mathrm{diff}}(\lambda)-3)\alpha_{\mathrm{dir}}(\lambda,\tilde{\theta}_s) + \mathrm{r}_{\mathrm{diff}}(\lambda,\theta_s)(1-\alpha_{\mathrm{diff}}(\lambda))^2\right)$$
$$+o(\varepsilon^2)$$
$$= K(1-\mathrm{r}_{\mathrm{diff}}(\lambda,\theta_s))\alpha_{\mathrm{dir}}(\lambda,\tilde{\theta}_s) + \mathrm{r}_{\mathrm{diff}}(\lambda,\theta_s)\alpha_{\mathrm{diff}}(\lambda) + \varepsilon^2\mathrm{P} + o(\varepsilon^2) \tag{B6}$$

where $P = (1-\alpha_{\mathrm{diff}}(\lambda))(\mathrm{r}_{\mathrm{diff}}(\lambda,\theta_s)(1-\alpha_{\mathrm{diff}}(\lambda)) - (1-\mathrm{r}_{\mathrm{diff}}(\lambda,\theta_s))\alpha_{\mathrm{dir}}(\lambda,\tilde{\theta}_s))$

Figure 12 shows that $\varepsilon^2 P$ can be neglected for slope angles lower than $10°$ which is the case in the study (see Fig. 4).



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





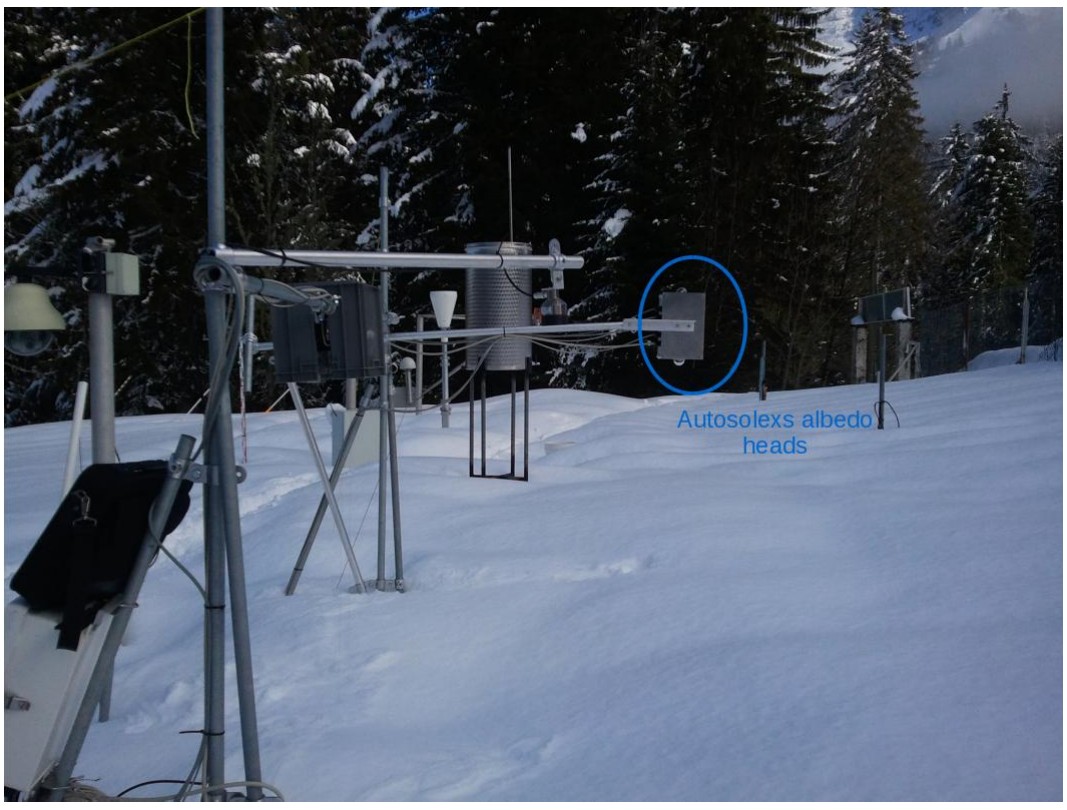

**Figure 1.** Part of the col de Porte field measurements site on 2014-02-17 14:35. North direction is the left of the picture. The two measurement heads of Autosolexs are circled in blue.





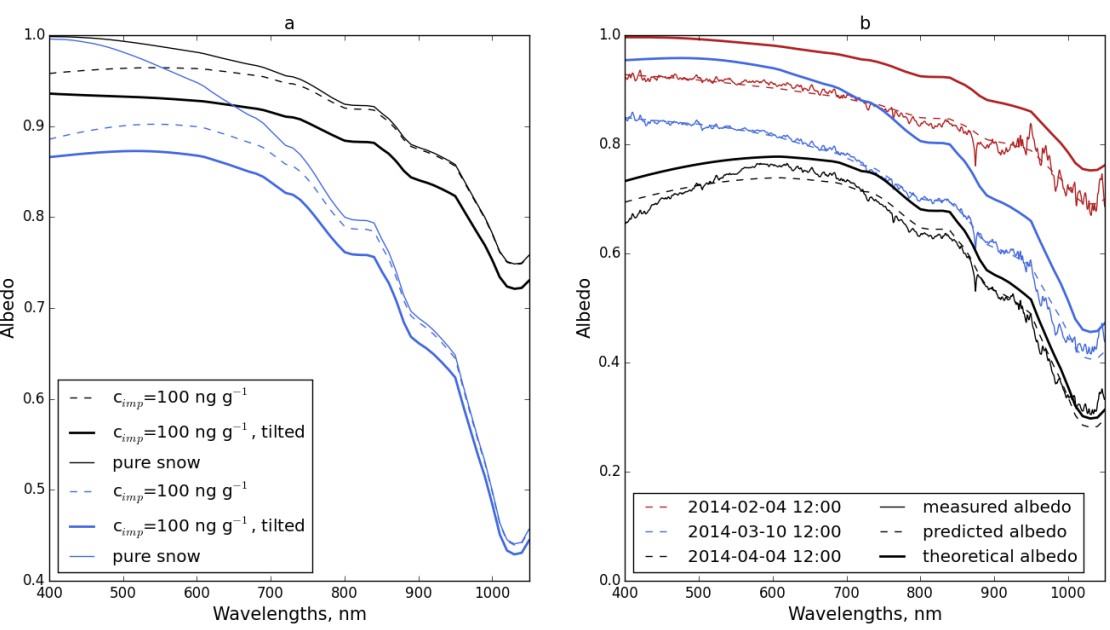

**Figure 2.** (a) Spectral albedo simulated using Eq. 6 for an horizontal surface and Eq. 8 for tilted surface ($\phi_n$=300 °, $\theta_n$=5 °) using estimated difuse to total irradiance ratio, $\phi_s$ and $\theta_s$ for 2014-03-08 at noon. The black lines correspond to SSA= 40 m$^2$ kg$^{-1}$ and the blue lines to SSA= 5 m$^2$ kg$^{-1}$. (b) Example of measured, predicted albedo and theoretical albedo on an horizontal surface for 3 dates in the season. For the red lines (resp. blue and black), optimal SSA is 36 m$^2$ kg$^{-1}$ (resp. 6.6, 2.6) and optimal c$_{\mathrm{imp}}$ is 0.5 ng g$^{-1}$ (resp. 15.1, 328). A is set to 0.943.



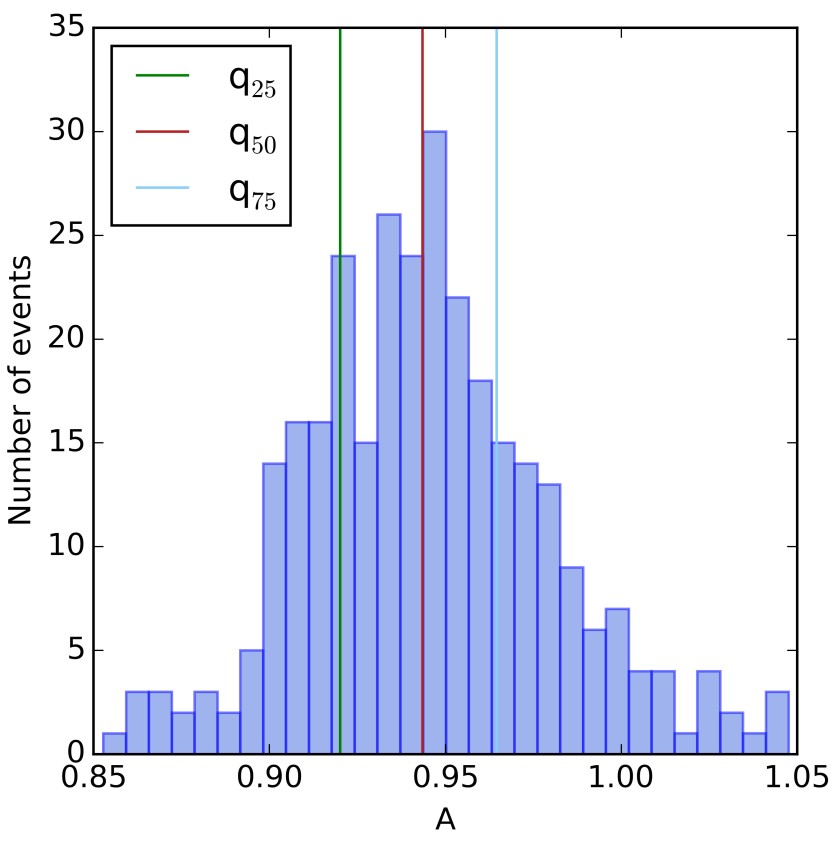

**Figure 3.** Distribution of the scaling factor, A for overcast days. Spectra used for this analysis corresponds to $\theta_s$ smaller than 65° and RMSD between predicted and measured spectrum lower than 0.02 over the 400-1050 nm range.





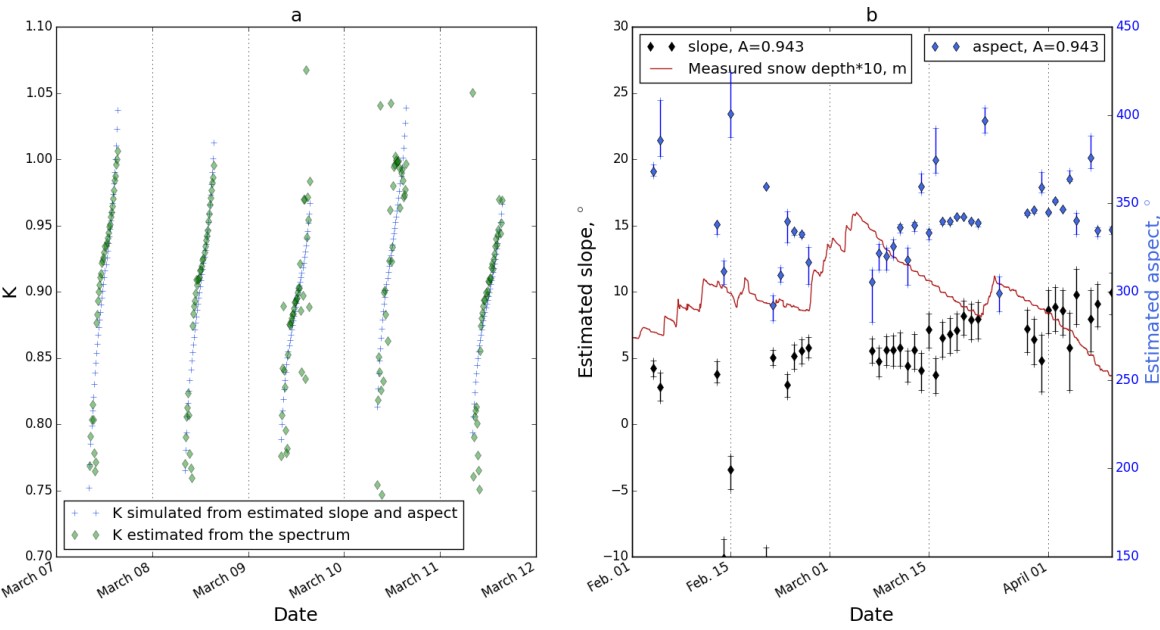

**Figure 4.** (a) Illustration of the determination of slope and aspect (*step 3*) for clear sky days. Green diamonds represent the diurnal cycle of parameter K retrieved after *step 2* and blue crosses show the simulated K using the estimated $\theta_n$ and $\phi_n$ from *step 3*. The scaling factor was set to A=0.943. (b) Optimal daily values of $\theta_n$ (in black) and $\phi_n$ (in blue) from *step 3*. Vertical error bars are obtained using quantiles 25 and 75 of the distribution of A. Daily values with range of aspect uncertainty larger than 80° are discarded. Spectra are also discarded if the RMSD between predicted and measured spectrum is lower than 0.022 over the 400-1050 nm range. Measured snow depth is shown in red.





**Figure 5.** Effect of liquid water presence on measured spectra (a) Example of two measured spectra on March 9th at 09:00 in blue and 13:12 in green and the associated wavelength of minimum reflectance represented by the vertical dotted lines. The vertical black line corresponds to $\lambda_{water}$. (b) Distribution of wavelength of minimum albedo in Eq. 9. Spectra are discarded if the RMSD between predicted and measured spectra is lower than 0.022 over the 400-1050 nm range. $\lambda_{water}$ is indicated by the black vertical line.





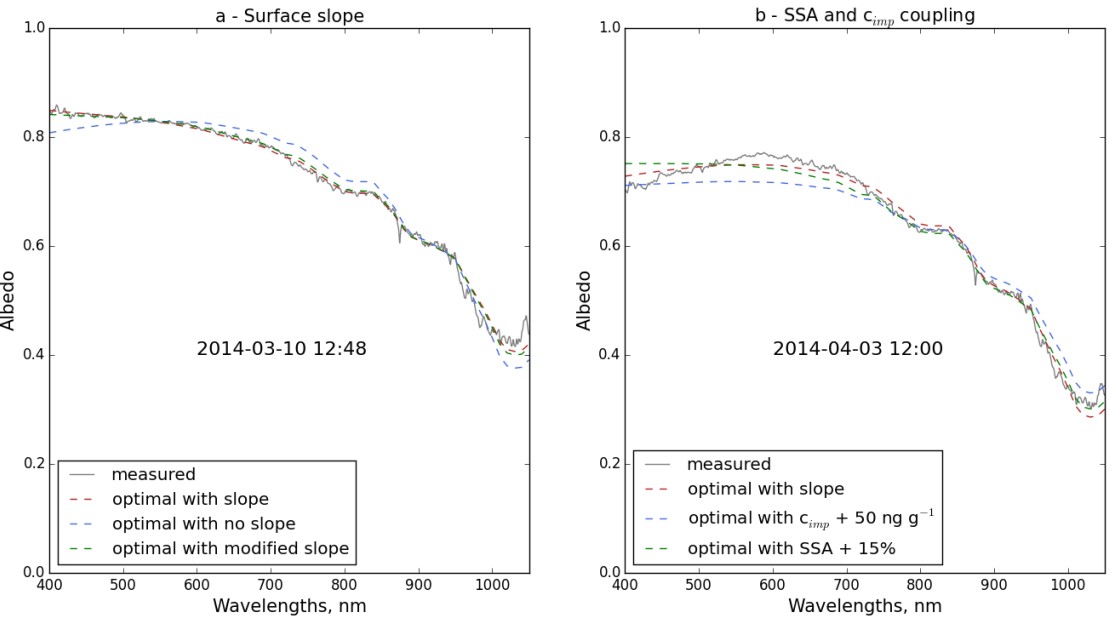

**Figure 6.** (a) Measured (black line) and optimal (dotted lines) spectra for 2014-03-10 12:48. The red line corresponds to *step 2*, optimal SSA (resp. $c_{imp}$) is 5.4 m$^2$ kg$^{-1}$ (resp. 16.2 ng g$^{-1}$). The blue line corresponds to the optimization considering that the surface is perfectly horizontal (SSA = 3.95 m$^2$ kg$^{-1}$, $c_{imp}$ = 129 ng g$^{-1}$). The green line correspond to the optimal spectrum with slope angles from *step 3* + 2° for the zenith angle and +10° for the azimuth (SSA = 5.0 m$^2$ kg$^{-1}$, $c_{imp}$ = 3.1 ng g$^{-1}$). (b) Measured (black line) and optimal (dotted lines) spectra for 2014-04-03 12:00. The red line corresponds to *step 2*, optimal SSA (resp. $c_{imp}$) is 2.82 m$^2$ kg$^{-1}$ (resp. 197.3 ng g$^{-1}$). The blue line corresponds to the optimization while adding 50 ng g$^{-1}$ to the optimal value of $c_{imp}$ (SSA = 3.35 m$^2$ kg$^{-1}$, $c_{imp}$ = 247.3 ng g$^{-1}$). The green line corresponds to adding 15 % to the optimal value of SSA (SSA = 3.24 m$^2$ kg$^{-1}$, $c_{imp}$ = 181.7 ng g$^{-1}$).





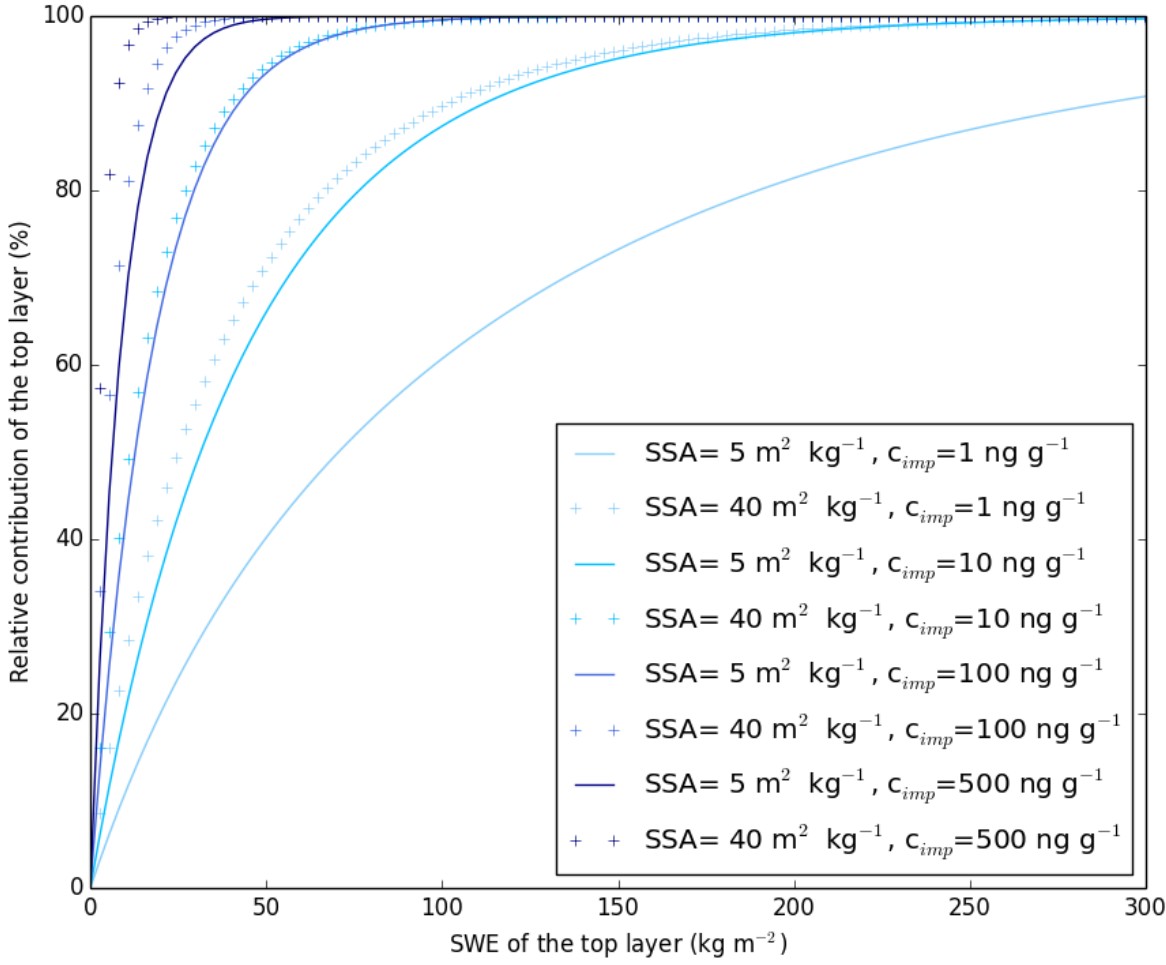

**Figure 7.** Relative contribution of the top layer of a semi-infinite snowpack to albedo averaged over 400-1050 nm as a function of top layer SWE ($h$) for low SSA (5 m$^2$ kg$^{-1}$, plain lines) and high SSA (40 m$^2$ kg$^{-1}$, dotted lines). The relative contribution is defined as $(\alpha(h) - \alpha(0))/(\alpha(\infty) - \alpha(0))$ where h is the SWE of the top layer of same SSA and c$_{\mathrm{imp}}$ modified by +10% with respect to the bottom layer. $\alpha(h)$ is the albedo of the snowpack constituted of these two layers.





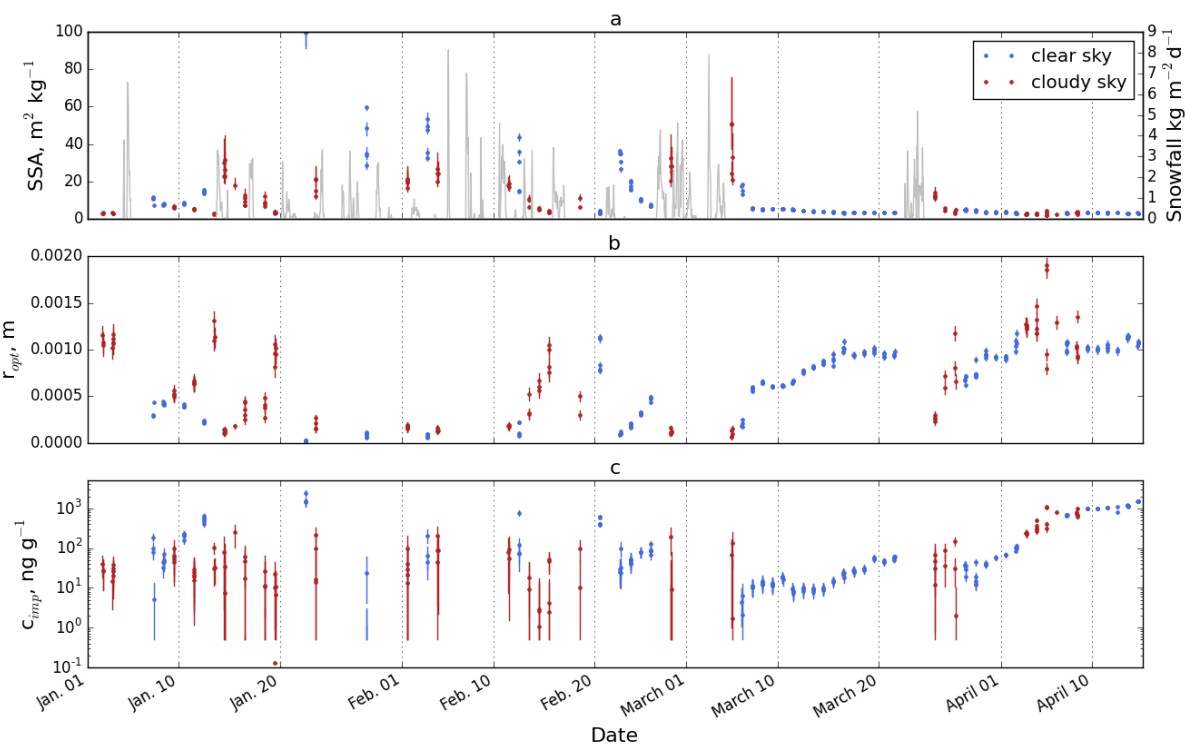

**Figure 8.** Seasonal evolution of near surface SSA (a), $r_{opt}$ (b) and $c_{imp}$ (c) predicted from measured spectra. Only spectra with RMSD between measured and predicted spectrum lower than 0.022 and measured between 12:00 and 13:00 UTC are indicated. Vertical error bars are obtained using quantiles 25 and 75 of A distribution. Clear sky days are indicated in blue and cloudy days in red. The measured snowfall rates are indicated in gray in panel (a).





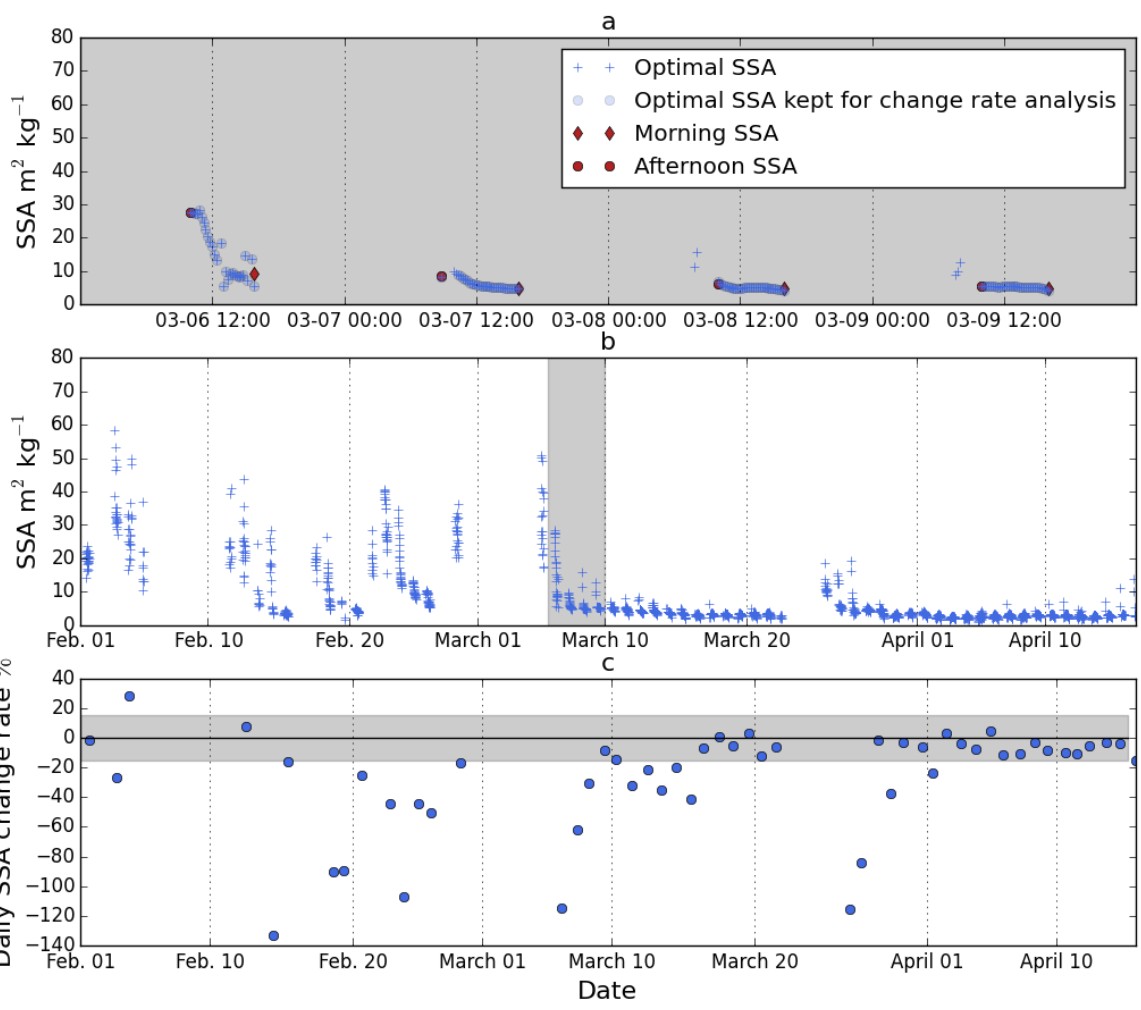

**Figure 9.** (a) Diurnal evolution of estimated near surface SSA between 2014-03-06 and 2014-03-09. Blue circles are the values kept for decay range analysis (panel c). The red diamonds (resp. circles) are the estimated morning (resp. afternoon) SSA values used to estimate the daily decay rate (panel c). (b) Seasonal evolution of estimate SSA for every spectra with RMSD lower than 0.022. (c) Estimated daily change rate in % of daily mean SSA values. The grey area corresponds to the 15% accuracy given in Picard et al. (2016a).





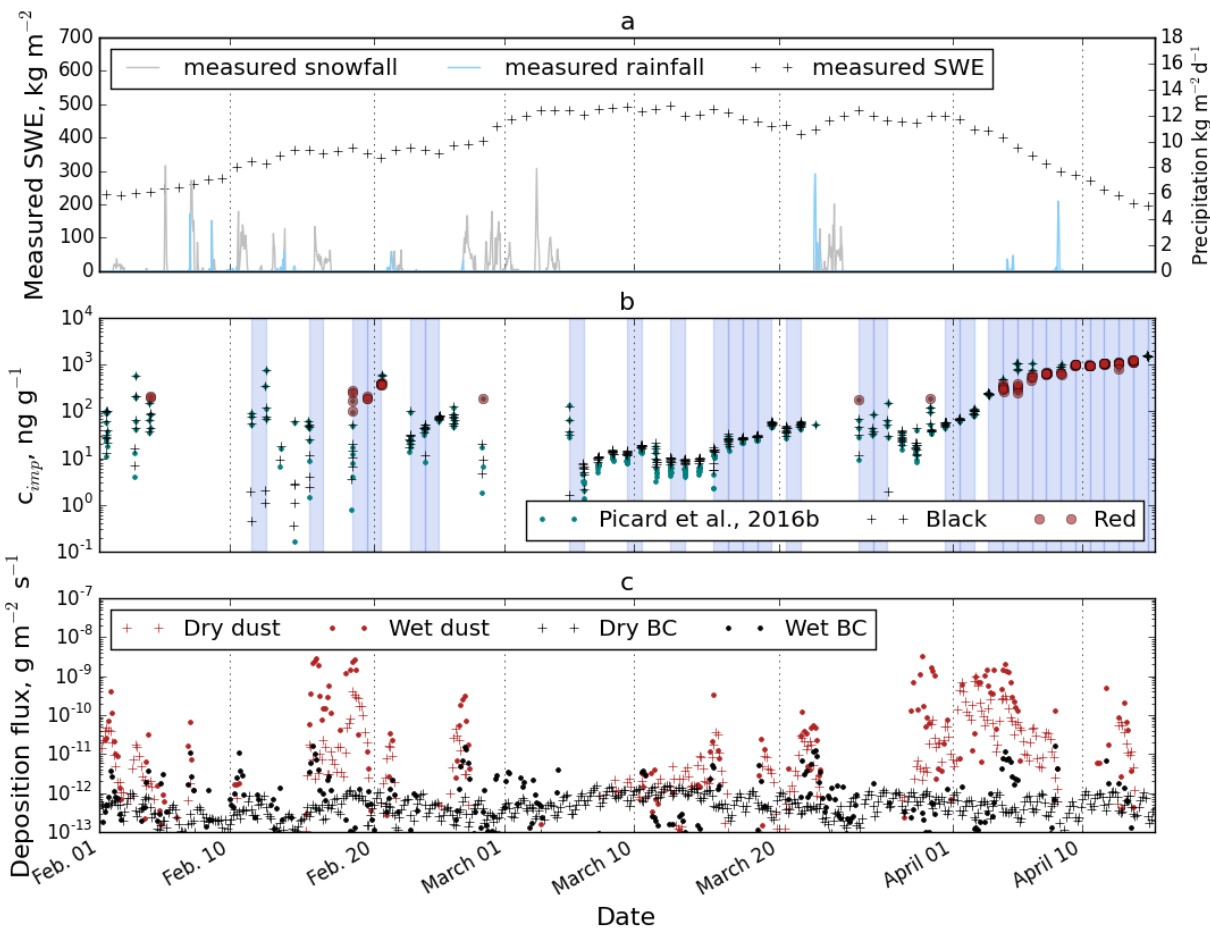

**Figure 10.** (a) Measured snow water equivalent (black cross), kg m$^{-2}$ and snow and rainfall rates (gray line) kg m$^{-2}$ d$^{-1}$ (b) near surface $c_{imp}$ estimated between 11:00 and 13:00. Red circle corresponds to $c_{imp}$ values for which the RMSD within 400-500 nm is larger than RMSD within 400-1050 nm. Only value with RMSD over 400-1050 lower than 0.022 are reported. The vertical blue bars indicated melting days estimated from measured SWE. Green dots are the values of near surface $c_{imp}$ estimated using the value of ice refractive index proposed in Picard et al. (2016b) instead of Warren and Brandt (2008). (c) Wet and dry deposition fluxes simulated by ALADIN-climate. Dusts are in red and black carbon in black.







**Figure 11.** (a) Wavelength of minimum reflectance as a function of measured volumetric liquid water content (LWC). (b) Measured surface temperature (black crosses). Spectra detected as wet are represented by the vertical blue bars and spectra detected as dry by the vertical yellow bar. Values presented in this figure are only for spectra with RMSD lower than 0.022. (c) Zoom of (b) between 2014-03-06 and 2014-03-09.





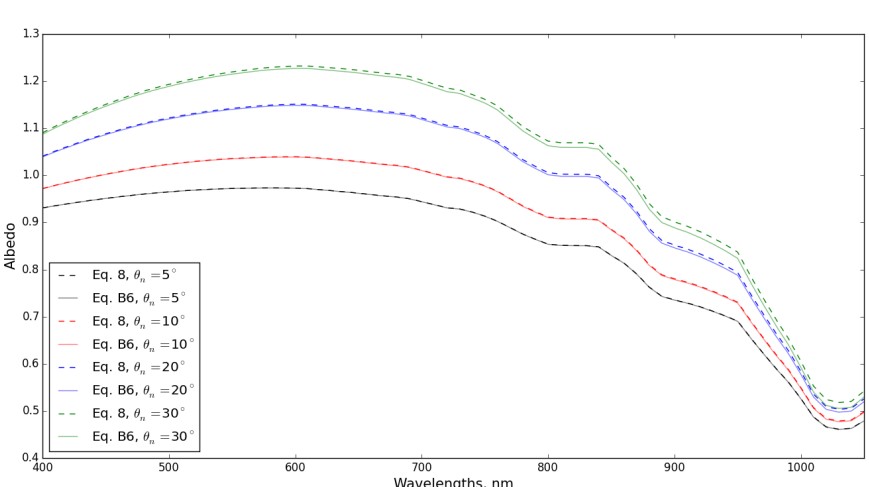

**Figure 12.** Comparaison of Eq. 8 (dotted lines) and Eq. B6 (solid lines) for the calculation of measured albedo for varying slope angles. The azimuth is set to $0°$ and the diffuse to total ratio and solar zenith angle are taken from 2014-03-07 noon data.