# Peer review of "In situ* continuous visible and near-infrared spectroscopy of an alpine snowpack"

_The Cryosphere, 2016_

## Referee Comment (RC1) · Anonymous Referee #1 · 22 Dec 2016

Review of

Marie Dumont, Laurent Arnaud, Ghislain Picard, Quentin Libois, Yves Lejeune, Pierre Nabat, Didier Voisin, and Samuel Morin. In situ continuous visible and near-infrared spectroscopy of an alpine snowpack; submitted to The Cryosphere Discuss., doi:10.5194/tc-2016-266, 2016

General comments:

This paper provides a very sound study on the determinations of several snow surface proprieties such as snow specific surface area (SSA), effective light-absorbing impurities content and presence of liquid water, based on spectral albedo measurements. The authors well describe the Methods and theoretical framework for analyzing the aforementioned effects depending on snow albedo. The paper also builds up on re-

[Figure]

Interactive
comment

cent studies in this field (e.g. Picard et al.). The methodological framework of handling the albedo data is quite elaborated including atmospheric model outputs and using several assumptions. I assume that these methodological framework is constructed to investigate the relationship between snow albedo and the snow cover proprieties.

To my opinion, the authors should analyze and present a bit more of the raw albedo data in order to explain the methods used. For me it is unclear why the authors are estimating the ratio of the diffuse and direct irradiance. To my knowledge, the albedo is defined as the ratio of the incoming and reflected global radiation. Both are affected by either direct sunlight or anisotropic reflectance depending on solar zenith angle (and cosine error of the entrance optics etc.). I assume that the authors take into account theses effects by using atmospheric model outputs and the methods describes in the paper.

However, it would be very useful and maybe simpler, to analyze the diurnal course of the snow albedo depending on solar zenith angle (SZA), when a minor change of the snow surface proprieties can be assumed. Most likely the spectral albedo shows a dependence on SZA – also depending on the wavelength selected, however, this dependency should be similar for all days of the season. If such a relationship can be found, all data of the days can be normalized to a reference SZA and then used for the comparison with the specific snow surface proprieties such as snow specific surface area (SSA), effective light-absorbing impurities content and presence of liquid water. This is in particular important for days with partly direct sun impact and partly cloudy in order to select either only direct sun albedo or cloud covered albedo. In this respect, it would be very useful the show the diurnal course of fully cloud albedo and to compare these effects with direct sun albedo. Due to this method, all SZA depending effects (e.g. cosine error of the entrance optics, slope of the surface) may be removed, which may allow to find a clearer signal between snow cover proprieties and spectral albedo.

In summary: I suggest to analyze and present some data of daily albedo depending on SZA and different cloud conditions and to discuss shortly these effects in regard

of SSA, LWC and impurities just based on the measurements before presenting the analysis including model calculations.

The submitted paper is well written and organized and the methods and data are fully described. The paper can be published with minor revision requested.

Specific comments:

Section 2.1: Some more details about the spectral albedo measurements are needed: - What is the height above snow surface of the two entrance optics, was the height constant over the season? Or was the height changing with growing snow cover? What would be the effect of the different hights. - What is the effect when the entrance optics where changes (from up to down and vice versa)? Was that tested? This would give a hint of the expected uncertainty of the albedo measurement, including all effects such a cosine error. - Spectral resolution of 3 nm: Do you mean the spectral bandwidth or the spectral sampling rate? - Are the domes heated?

Page 4, line 4: (iv) correction of the angular response: How did this correction applied. Was the correction applied for both entrance optics? Are the entrance optics similar in respect of the cosine error?

Page 6, line 18; Can the reflected radiation on snow be assumed as isotropic. Is there any reference for this assumption? Most likely the reflected radiation also shows a dependency on SZA.

Page 7: line 6. Maybe there is a strong dependence of the scaling factor depending on SZA. Was this analyzed? Maybe this explains the distribution of the scaling factor A in Figure 3.

Smaller issues:

The abstract basically describes the intention and results of the study – no changes.

The text is well written and no major typos have been detected so far..

---

## Referee Comment (RC2) · Anonymous Referee #2 · 8 Jan 2017

General comments

The paper presents continuous snow spectral albedo observations over the range 350-1050 nm from an Alpine area. The dataset is extremely valuable and unique, as very few continuous spectral albedo datasets have been obtained so far in the world. The dataset allowed the examination of the diurnal and seasonal evolution of some snowpack properties that affect the albedo (namely SSA, snow impurity content, surface slope, and presence of liquid water) and that can be retrieved from the albedo data thought the inversion of snow albedo schemes. This ambitious goal is of high relevance for remote sensing applications.

The dataset is extremely complex, with uncertainties arising from a large number of sources. The authors have well taken into account this complexity, and have developed an elaborated method to retrieve the snow properties from the albedo spectra. The study is very innovative, and opens the path for a better exploitation of albedo observations. However, in my opinion the presented methodology is not illustrated clearly enough. It is sometimes hard to follow the reasoning behind the proposed method, or even understand what exactly the method does, and consequently it is difficult to interpret the results. In several parts the text is too cryptic and condensed. A better readability of the paper is a necessary condition for the adoption of the proposed methodology by the scientific community.

I therefore recommend the authors to do a major revision, which should mainly consist in reformulating most of the text in the method section to better clarify the content and provide all the necessary details to help the reader to easily follow the reasoning behind the various steps.

Detailed comments

p.5, line 5 "...thus impose the observed unfolding of the meteorological conditions that is essential to represent dust event". This is an example of too cryptic and condensed text. Could you please rewrite to make the content clearer?

p.5 Sect. 3.1. The content of this section should be formulated much more clearly, and with a more strict logic. In my understanding the following steps were done: 1. ALADIN-Climate forced by Era-Interim data was applied to calculate optically relevant atmospheric quantities (aerosol, ozone, water vapor) in the 50x50 km grid cell that includes Col de Porte, for the whole observational period. Did the authors used a single model cell or made some sort of weighted average? The authors write that the model was used to calculate "mean atmospheric conditions at Col de Porte" (line 16). Does this imply some averaging in time or space was done? Please clarify. The atmospheric profiles obtained from the ALADIN-Climate runs were considered representative of Col de Porte. This is not an obvious passage, as in a mountain area the 50x50 km resolution can be too low and strong differences can exist inside the grid. This problem clearly

appears in Fig 10, but the authors should discuss this issue already in the methodology, and explain the limitations and benefits of the chosen approach. 2. The obtained atmospheric profiles and the locally measured T2m were fed into SBDART to calculate the ratio between Swdir and SwTOT as a function of the cloud optical thickness $\tau$ (for the whole observation period?). These results were used to produce a regression equation of $\tau$ vs Swdir/SwTOT. 3. The regression equation was applied to calculate the actual $\tau$ from the observed Swdir and SwTOT. 4. SBDART was then applied again to compute hourly direct and diffuse irradiance using the derived $\tau$. Is my interpretation correct? If it is so, please describe these steps in this logical sequence. Also, how well the Swdir and SwTOT modelled with SBDART matched the observations? In other words, what is the uncertainty in Eq 1, and how does it propagate to the calculation of the Swdir and Swdiff spectra?

p.6, lines 19-20 "the surface slope is small and local enough not to modify significantly the solid angles under which the incoming and reflected radiations are measured with respect to what would happen for an horizontal surface" Very tortuous sentence, quite difficult for me to understand. Can it be made clearer? Sect. 3.3 is quite hard to follow. The equations to calculate the effect of slope on diffuse radiation and on measured albedo are presented in Appendices A and B, but without explaining many passages, so it is too laborious for me to check them, and it is difficult for an interested reader to apply them without a full understanding. The authors refer to some literature for the details (Dumont et al, 2011, Wang et al 2016), but they need to report in the paper the key concepts and passages to make the paper self-sufficient. For instance, what is the physical meaning of the parameter "K"? Probably also a schematic drawing of the angles of the tilted and horizontal surface would help to understand the equations.

p. 7, line 5: "…and an horizontal surface". Should instead be a "tilted surface"? At the beginning of Sect. 3.3.1. the authors could add a paragraph introducing the strategy applied in the method and the purpose of the various steps.

p.7, line 17: "A seasonal value of A is estimated…" Why do the authors need to calculate a seasonal value? I can understand it after seeing Fig. 3, but this figure is introduced only in the Result section, so here the authors need to explain why A can vary and why they need to choose a single value. The reader can then better understand the sentence about the propagation of errors (lines 21-22).

p.7, line 19: "To avoid undetermination problem between A and Cimp. . .". This is a too cryptic and compact sentence, please explain.

p. 7, lines 27-30: I don't understand this paragraph. First of all, how the (albedo?) spectra are calculated (line 27)? I suppose using the retrieved SSA and Cimp. And which model was applied? If Cimp is retrieved with the optimization method, why the discrepancy between observed and calculated albedo (using the retrieved optimal parameters?) should be related to Cimp? In my view, the discrepancy can be attributed to a large number of approximations and assumptions in the applied model.

p.8, lines 4-6: I recommend the authors to explain the motivation of Step 3, which looks unnecessary if they know in advance the slope and aspect of the surface. Also, I would explain here why the slope can change (with precipitation and snowdrift).

p.8, lines 9-10: How is the filtering of the spectra done? Do the authors remove the spectra that show a specific features in the moving window average?

p.8, lines 18: ". . .only slightly lower than the ideal value of one". This is not a correct expression, as in my understanding A=1 is not an "ideal value" of A, but rather it indicates that the instrument had an ideal response. So, the concept would better be expressed by stating that a value of A close to 1 indicates that the measurement apparatus has caused only a small deviation from the ideal response of the instrument.

p.8, line 20: please remove "presence of"

p.8, lines 25-26: ". . . diurnal cycle with higher values in the morning and lower values in the afternoon". From Fig 4a I see that K is higher in the afternoon! Please consider redrawing Fig 4a, with just 24 hours in the x-axis to show better the diurnal cycle, and

maybe the time series of K in just two selected days, one with good agreement and one will poor agreement.

In Fig. 4b, please replace the symbol "âǓę" with "(degrees)" in the y-labels for slope and aspect.

p. 9, line 2: "The seasonal evolution of slope and aspect seems to be related to snow evolution". It has to be, what else could cause a change in slope and aspect? So, I would replace "seems to be" with "is".

p. 9, line 21: "have been done under" should perhaps be "have been obtained under".

p. 11, lines 14-16: The text is too difficult to follow. The authors need to provide some interpretation, and also explain the applied relationship between SWE and snow density.

p. 12, line 1: "2014-04-10" should perhaps be "2014-04-02"?

p.12, line 8-9: "The grey area corresponds to the 15% uncertainty estimated in Picard et al. (2016a)". Please clarify what this uncertainty refers to. In Fig 9c it appears as a constant value throughout the observational period rather than a percentage of SSA (which varied in the range ∼5-60 m2/kg).

p.12, lines 24-29: I did not understand this paragraph. The authors write "If the RMSD is larger than the RMSD over the whole spectrum, Cimp is represented in red". What is the relevance of this? And what is the reasoning behind the application of this criterion? The message of this paragraph is totally unclear to me.

Fig 9: In panel (a), I think that morning and afternoon symbols are inverted.

[Figure]

---

## Short Comment (SC1) · 9 Jan 2017

The authors measured snow spectral albedo in the visible/NIR range at an alpine site and highlighted the effects of snow specific surface area, impurity content, presence of liquid water, and slope on variations of spectral snow albedo. I have a short comment.

In addition to the factors mentioned by the authors, recent studies also showed that snow grain shape and how impurities mixed with snow grains are two critical factors in determining snow albedo (e.g., Liou et al., 2014; He et al., 2014). I suggest including these references and adding some discussions on this aspect, which would be very interesting.

References:

[Figure]

Liou, K. N., Takano, Y., He, C., Yang, P., Leung, L. R., Gu, Y., and Lee, W. L.: Stochastic parameterization for light absorption by internally mixed BC/dust in snow grains for application to climate models, J. Geophys. Res.-Atmos., 119, 7616-7632, doi:10.1002/2014jd021665, 2014.

He, C., Li, Q. B., Liou, K. N., Takano, Y., Gu, Y., Qi, L., Mao, Y. H., and Leung, L. R.: Black carbon radiative forcing over the Tibetan Plateau, Geophys. Res. Lett., 41, 7806-7813, doi:10.1002/2014gl062191, 2014.

---

## Author Comment (AC1) · 10 Feb 2017

Authors responses are enlighten in blue. Proposed changes in the manuscript are reported in bold.

**General comments:**
This paper provides a very sound study on the determinations of several snow surface proprieties such as snow specific surface area (SSA), effective light-absorbing impurities content and presence of liquid water, based on spectral albedo measurements. The authors well describe the Methods and theoretical framework for analyzing the aforementioned effects depending on snow albedo. The paper also builds up on recent studies in this field (e.g. Picard et al.). The methodological framework of handling the albedo data is quite elaborated including atmospheric model outputs and using several assumptions. I assume that these methodological framework is constructed to investigate the relationship between snow albedo and the snow cover proprieties.

The authors are thankful for this useful review of the manuscript. All comments have been accounted for, responses and proposed modifications are described below after each comment.

To my opinion, the authors should analyze and present a bit more of the raw albedo data in order to explain the methods used. For me it is unclear why the authors are estimating the ratio of the diffuse and direct irradiance. To my knowledge, the albedo is defined as the ratio of the incoming and reflected global radiation. Both are affected by either direct sunlight or anisotropic reflectance depending on solar zenith angle (and cosine error of the entrance optics etc.). I assume that the authors take into account theses effects by using atmospheric model outputs and the methods describes in the paper.

The direct to total spectral irradiance ratio is required to perform the cosine response correction. It is also required to analyse the snow surface parameters and the effect of slope on the albedo measurement. The effects of the cosine receptor angular response and of SZA are taken into account in the methodology. The effects of the anisotropy of reflected and diffuse incoming radiations are neglected (see details in the response to the specific comment 3).

However, it would be very useful and maybe simpler, to analyze the diurnal course of the snow albedo depending on solar zenith angle (SZA), when a minor change of the snow surface proprieties can be assumed. Most likely the spectral albedo shows a dependence on SZA – also depending on the wavelength selected, however, this dependency should be similar for all days of the season. If such a relationship can be found, all data of the days can be normalized to a reference SZA and then used for the comparison with the specific snow surface proprieties such as snow specific surface area (SSA), effective light-absorbing impurities content and presence of liquid water. This is in particular important for days with partly direct sun impact and partly cloudy in order to select either only direct sun albedo or cloud covered albedo. In this respect, it would be very useful the show the diurnal course of fully cloud albedo and to compare these effects with direct sun albedo. Due to this method, all SZA depending effects (e.g. cosine error of the entrance optics, slope of the surface) may be removed, which may allow to find a clearer signal between snow cover proprieties and spectral albedo.

In summary: I suggest to analyze and present some data of daily albedo depending on SZA and

different cloud conditions and to discuss shortly these effects in regard of SSA, LWC and impurities just based on the measurements before presenting the analysis including model calculations.

Thanks for these thoughts on the methodology. "Most likely the spectral albedo shows a dependence on SZA – also depending on the wavelength selected, however, this dependency should be similar for all days of the season." The dependency on the SZA (ignoring the effect of snow properties) is not exactly similar for all days of the season because of the changes in the diffuse to total irradiance ratio, i.e. changes in atmospheric profiles and cloudiness and also because of the changes in the slope and aspect of the surface (due to precipitation, wind transportation and melt).
In response to this comment, we have added the figure below that shows some examples of the raw albedo diurnal cycle for a cloudy day and a clear sky day.

[Figure]

**Figure 3 : Raw measured albedo for a clear sky day (blue lines, 2014-04-03) and for a cloudy day (red lines, 2014-02-01) at 10:00 (dashed lines), 12:00 (solid lines) and 14:00 (dash-dotted lines).**

A short description of the Figure has been added page 7 line 12.

**Figure 3 illustrates the raw measured albedo diurnal cycles for a cloudy day (red lines) and for a clear sky day (blue lines). As expected from Eq. 8, the diurnal cycle is more pronounced for the clear sky day, the albedo evolution being non-symmetric with respect to solar noon probably due to both slope and changes in snow properties effects.**

The submitted paper is well written and organized and the methods and data are fully described. The paper can be published with minor revision requested.

**Specific comments:**

**1** - Section 2.1: Some more details about the spectral albedo measurements are needed:

Agree. The details listed below in blue have been added in section 2.1. (see below)

 - What is the height above snow surface of the two entrance optics, was the height constant over the season? Or was the height changing with growing snow cover? What would be the effect of the different hights.-

The height of the upward looking head is  2.4 m above the ground (with no snow) and 2.1 m for the downward looking head.  The height is changing with the growing snow cover, the main effect of this varying height is the change in the surface seen by the sensors.

These details have now been added in the text page 3 line 28 :
"one albedo head. **The upward (downward) optic is set up 2.4 (2.1) m above bare soil. The height of the albedo heads and consequently, the field of view of the sensor, are thus varying with snow depth.** The device …"

What is the effect when the entrance optics where changes (from up to down and vice versa)? Was that tested? This would give a hint of the expected uncertainty of the albedo measurement, including all effects such a cosine error.

The entrance optics are cross-calibrated and this is taken into account in the collector cosine response correction (please see response to comment 2 for details and proposed modifications in the paper).

- Spectral resolution of 3 nm: Do you mean the spectral bandwidth or the spectral sampling rate?

 We mean spectral bandwidth.
Page 3 line 29 has thus been modified as follows:
"350-1050 nm range with an effective spectral resolution**, i.e. spectral bandwidth,** of 3 nm."

- Are the domes heated?
Yes, and ventilated
This detail has been added page 3 line 31 : "Consequently, the device was cleaned up manually after each snowfall **although both the upward and downward looking domes are heated and ventilated**. The snow surface … "

**2** - Page 4, line 4: (iv) correction of the angular response: How did this correction applied. Was the correction applied for both entrance optics? Are the entrance optics similar in respect of the cosine error?

As stated in the text (page 4 lines 3-5), the correction of the collector angular response is performed exactly as extensively detailed in Picard et al., 2016a (section 3.3.4). The correction is applied only to the direct component as the diffuse component has already been calibrated at the cross calibration step. The correction is thus applied only to the upward looking entrance optics. The cross calibration step consists in measuring successively the upward and downward channels under the same illumination conditions (see section 3.3.3 in Picard et al., 2016a) and allows accounting for the differences in the two entrance optics cosine errors. The cosine errors of the two entrance optics are of the same order of magnitude.

The text of the paper was consequently modified as follows page 4 line 4 :

'...(iv) collector angular response. **The correction of the collector angular response was applied in two steps (i) cross-calibration of the two entrance optics under the same illumination conditions and (ii) cosine response correction on the direct component of the incident radiation as detailed in Picard et al., 2016a (sections 3.3.3 and 3.3.4).** The corrected spectra ....'

**3** - Page 6, line 18; Can the reflected radiation on snow be assumed as isotropic. Is there any reference for this assumption? Most likely the reflected radiation also shows a dependency on SZA.

The radiation reflected by the snowpack is not isotropic (e.g. Dumont et al., 2010) and the anisotropy of the reflected radiations indeed varies with SZA: the anisotropy is stronger for higher SZA. The anisotropy of the reflected radiation impacts the cosine correction as detailed in Carmagnola et al., 2014. The effect of this anisotropy on the spectra correction is of second order as long as the cosine response correction is small, which is the case for our device.

This is discussed page 7 lines 8-12 and also in the conclusion page 14 line 3.

**4** - Page 7: line 6. Maybe there is a strong dependence of the scaling factor depending on SZA. Was this analyzed? Maybe this explains the distribution of the scaling factor A in Figure 3.

The distribution of scaling factor A in Figure 3 was obtained during fully cloudy days, the SZA should not have much impact on the spread. We didn't analysed in detail how A can varies with SZA, indeed we assume that most of the variations of the signal with SZA are due to slope and albedo effect (Eq. 8). Due to the number of unknowns in Eq. 8, it is quite difficult to disentangle the effect of SZA on A, K and albedo.
To account for the referee comments we modified the text page 8 lines 1-2:
"Note that by using a seasonal A value for every spectra, we assume that the measurements artefacts are the same under cloudy and clear sky conditions **and for varying solar zenith angles.**"

**Smaller issues:**
The abstract basically describes the intention and results of the study – no changes.
The text is well written and no major typos have been detected so far..

Thanks !

---

## Author Comment (AC2) · 10 Feb 2017

Authors responses are enlighten in blue. Proposed changes in the manuscript are reported in bold.

**General comments**

The paper presents continuous snow spectral albedo observations over the range 350-1050 nm from an Alpine area. The dataset is extremely valuable and unique, as very few continuous spectral albedo datasets have been obtained so far in the world. The dataset allowed the examination of the diurnal and seasonal evolution of some snowpack properties that affect the albedo (namely SSA, snow impurity content, surface slope, and presence of liquid water) and that can be retrieved from the albedo data thought the inversion of snow albedo schemes. This ambitious goal is of high relevance for remote sensing applications.

The dataset is extremely complex, with uncertainties arising from a large number of sources. The authors have well taken into account this complexity, and have developed an elaborated method to retrieve the snow properties from the albedo spectra. The study is very innovative, and opens the path for a better exploitation of albedo observations. However, in my opinion the presented methodology is not illustrated clearly enough. It is sometimes hard to follow the reasoning behind the proposed method, or even understand what exactly the method does, and consequently it is difficult to interpret the results. In several parts the text is too cryptic and condensed. A better readability of the paper is a necessary condition for the adoption of the proposed methodology by the scientific community.

I therefore recommend the authors to do a major revision, which should mainly consist in reformulating most of the text in the method section to better clarify the content and provide all the necessary details to help the reader to easily follow the reasoning behind the various steps.

The authors are grateful to the referee for this in-depth and useful review of the manuscript. All comments have been accounted for and the manuscript has been modified as detailed below after each referee comment. The methodology description and the writing have been clarified as recommended by the referee. The authors hope that the new version of the manuscript is easier to follow and clearer.

**Detailed comments**

**1** - p.5, line 5 ": : :thus impose the observed unfolding of the meteorological conditions that is essential to represent dust event". This is an example of too cryptic and condensed text. Could you please rewrite to make the content clearer?

Agree. The sentence was reformulated as follows:
"Even if ALADIN-Climate is a regional climate model, the model has the ability to reproduce the observed weather chronology thanks to the spectral nudging method (Radu et al., 2008), which enables us to keep the large scales atmospheric conditions from the boundary forcing. **The accuracy of the chronology of meteorological episodes is indeed essential to correctly represent the chronology of aerosols deposition on the snowpack**. "

**2** - p.5 Sect. 3.1. The content of this section should be formulated much more clearly, and with a more strict logic. In my understanding the following steps were done: 1. ALADIN-Climate forced by Era-Interim data was applied to calculate optically relevant atmospheric quantities (aerosol, ozone, water vapor) in the 50x50 km grid cell that includes Col de Porte, for the whole observational period. Did the authors used a single model cell or made some sort of weighted average? The authors write that the model was used to calculate "mean atmospheric conditions at Col de Porte" (line 16). Does this imply some averaging in time or space was done? Please clarify. The atmospheric profiles obtained from the ALADIN-Climate runs were considered representative of Col de Porte. This is not an obvious passage, as in a mountain area the 50x50 km resolution can be too low and strong differences can exist inside the grid. This problem clearly appears in Fig 10, but the authors should discuss this issue already in the methodology, and explain the limitations and benefits of the chosen approach. 2. The obtained atmospheric profiles and the locally measured T2m were fed into SBDART to calculate the ratio between Swdir and SwTOT as a function of the cloud optical thickness _ (for the whole observation period?). These results were used to produce a regression equation of _ vs Swdir/SwTOT. 3. The regression equation was applied to calculate the actual _ from the observed Swdir and SwTOT. 4. SBDART was then applied again to compute hourly direct and diffuse irradiance using the derived _ . Is my interpretation correct? If it is so, please describe these steps in this logical sequence. Also, how well the Swdir and SwTOT modelled with SBDART matched the observations? In other words, what is the uncertainty in Eq 1, and how does it propagate to the calculation of the Swdir and Swdiff spectra?

Thanks a lot for this comment. We agree that the section was not sufficiently logically described and yes the interpretation of the referee is correct. The section was largely rewritten and details were added in section 2.3 as follows:

Page 5 line 8 : " … dry deposition fluxes. **In this study, we used the outputs of a single model cell, the closest to Col de Porte site. The horizontal distance of the model cell center to the site is 22.6 km and the grid cell is located 800 m below Col de Porte site."**

Page 5 line 10 :
"3.1 Estimation of the direct to diffuse solar irradiance ratio

The ratio of diffuse over direct irradiance is required to perform accurate correction of the measured spectrum (e.g. Picard et al., 2016a). **The calculation of such ratio requires the knowledge of the atmospheric profiles and of the local cloud optical thickness, τ.**
**We thus estimate this local cloud optical thickness using both the atmospheric profiles from ALADIN-Climate and local meteorological observations at Col de Porte (Morin et al., 2012) to overcome the coarse resolution of ALADIN-Climate. The first step consists in estimating a relationship between τ and the broadband direct (SWdir) to total (SWtot) ratio.** For this purpose, the SBDART detailed radiative model (Ricchiazzi et al., 1998) was used to calculate SWdir over SWtot from varying cloud optical thicknesses and mean atmospheric conditions at Col de Porte (aerosols optical thickness, total ozone column and total water vapour column). **The mean atmospheric conditions were derived from ALADIN-Climate outputs and the measured 2m air temperature average over the measurements period.** A regression equation (Eq. 1) was derived from those results.

**As a second step, Eq. 1 was** used to estimate τ from SWdir over SWtot ratio measured at Col de Porte. "
**Finally**, the outputs of the ALADIN-Climate …. analysis. **Note that the broadband SWdir and SWtot estimated from Col de Porte measurements and simulated with SBDART agree within ±10 Wm⁻². The accuracy of the simulated spectral direct to diffuse solar irradiance ratio has not been**

**evaluated in absence of measurements. The difference in elevation between the ALADIN-Climate grid cell and Col de Porte site might lead to an overestimation of the diffuse fraction in the visible wavelengths.** ”

The limitations due to the coarse resolution of ALADIN-Climate have also been underlined in the concluding remarks.
 Page 14 line 3
“… future work. **Further refinements on the calculation of the spectral direct to total irradiance ratio or simultaneous measurements of this ratio and albedo are required to improve the accuracy of the method.** ”

**3** - p.6, lines 19-20 “the surface slope is small and local enough not to modify significantly the solid angles under which the incoming and reflected radiations are measured with respect to what would happen for an horizontal surface” Very tortuous sentence, quite difficult for me to understand. Can it be made clearer?

Agree, the whole paragraph has been modified as follows:
“This change is of crucial importance for our application since it is wavelength-dependant. **In addition to the change in the effective sun incident angles, the surface slope modifies (i) the solid angle under which the sky is viewed from the surface and thus the incoming amount of diffuse solar radiation and (ii) the total amount of radiation received by the snow surface since it receives some of the radiation reflected by the adjacent slopes. The upward radiation measured by the horizontal sensor is also modified with respect to what would happen if the sensor were parallel to the surface since part of the field of view sees the atmosphere and not the snow surface. In the following,** we assume that (i)   ”

Sect. 3.3 is quite hard to follow. The equations to calculate the effect of slope on diffuse radiation and on measured albedo are presented in Appendices A and B, but without explaining many passages, so it is too laborious for me to check them, and it is difficult for an interested reader to apply them without a full understanding. The authors refer to some literature for the details (Dumont et al, 2011, Wang et al 2016), but they need to report in the paper the key concepts and passages to make the paper self-sufficient. For instance, what is the physical meaning of the parameter "K"? Probably also a schematic drawing of the angles of the tilted and horizontal surface would help to understand the equations.

The parameter K is the relative change in the cosine of the sun effective incident angle due to the local tilt of the snow surface.
This has been added  page 7 line 3.
“(e.g. Dumont et al., 2011) where **K is the relative change in the cosine of the sun effective incident angle to the local tilt of the snow surface and writes »**
The schematic drawing below has also been added in the paper so that equations in Appendices A and B can be understood more clearly.

[Figure]

"Figure 2. Schematic drawing of the tilted snow surface (blue plane) and associated angles. The grey plane corresponds to the horizontal plane. The black reference frame is the one attached to the tilted surface. The grey arrow represents the vertical with respect to the horizontal plane. H(Φ) is the horizon line in the tilted reference frame. Tilde(Θs) and tilde(Φs) are the zenith and azimuth sun angles in the tilted reference frame. θn is the slope elevation and Φn the aspect. "

References to this figure has been added page 7 line 2 and page 15 line 3.

Several details have also been added in Appendix B.
Page 15 line 23 :
"The first (resp. second) term of the sum in Eq. B2 corresponds to the incoming direct (resp. diffuse) radiation on the tilted surface. The last term of the sum is the amount of radiation incoming on the surface and reflected by adjacent slope."
Page 16 line 3 :
"The first term of the sum in Eq. B4 corresponds to the amount of radiation reflected by the snow surface and seen by the sensor. The second term is the amount of diffuse atmospheric radiation seen by the sensor."

**4** - p. 7, line 5: ": : :and an horizontal surface". Should instead be a "tilted surface"? At the beginning of Sect. 3.3.1. the authors could add a paragraph introducing the strategy applied in the method and the purpose of the various steps.

P7 line 5 has been modified accordingly, thanks for noticing this typo.
For the second part of the comment, please refer to response to comment 6 and the new introductory paragraph of section 3.3.1.

**5** - p.7, line 17: "A seasonal value of A is estimated: : :" Why do the authors need to calculate a seasonal value? I can understand it after seeing Fig. 3, but this figure is introduced only in the Result section, so here the authors need to explain why A can vary and why they need to choose a single value. The reader can then better understand the sentence about the propagation of errors (lines 21-22).

Please see response to referee comment 6 below.

**6** - p.7, line 19: "To avoid undetermination problem between A and Cimp: : :". This is a too cryptic and compact sentence, please explain.

The authors agree that the whole sentence is not easily understandable. On the contrary to the choice made in Picard et al., 2016a to estimate one value of A for each spectrum, in the study we chose to estimate only one value of A for the whole season. This choice was led by the fact that on the contrary to the snow at Dome Concordia, snow at Col de Porte may contain light absorbing impurities in sufficient concentration to affect the albedo. Eq. 8 contains 4 unknowns in our case, A, SSA, C*imp* and K. For moderate to high amount of impurities, several A and Cimp values can lead to approximately the same modelled spectrum. That's why we made the choice to use only one value of scaling factor A for the whole season. We have modified section 3.3.1 as follows:

"In order to relate variations of spectral albedo to variations of surface snow properties, we apply the following methodology to the measured albedo. **The main idea of the methodology is to use fit Eq. 8 using optimal parameters for each spectrum. Eq. 8 indeed contains 4 unknowns namely A, SSA, Cimp and K. For moderate to high amount of impurities in snow, several (A, Cimp, SSA) triplets can lead to approximately the same modelled spectrum. For this reason, we chose to first set the scaling factor A to a constant value for the whole season (*Step 1*). Optimal SSA, Cimp and K are then estimated for each spectrum (*Step 2*). The diurnal cycles of parameter K are then used to estimate a daily value for surface slope and aspect (*Step 3*). Although this step is not required for the estimation of optimal SSA and Cimp, it provides a further verification of the physical consistency of the methodology. Finally, the measured spectra are analysed with respect to the presence of liquid water (*Step 4*).**

*Step 1 : Estimate the scaling factor.* (…)"

**7** - p. 7, lines 27-30: I don't understand this paragraph. First of all, how the (albedo?) spectra are calculated (line 27)? I suppose using the retrieved SSA and Cimp. And which model was applied? If Cimp is retrieved with the optimization method, why the discrepancy between observed and calculated albedo (using the retrieved optimal parameters?) should be related to Cimp? In my view, the discrepancy can be attributed to a large number of approximations and assumptions in the applied model.

The albedo is calculated using optimal SSA and Cimp and Eq. 8. (predicted albedo in Figure 2b). As explained in Section 3.2 the impurities effect of the albedo is modelled using black carbon only (which has a flat spectral signature). Other impurities such as mineral dust can be found in the snow and in the Alps, mineral dust often has a reddish signature. Consequently, while estimating the optimal spectrum, the reddish signature of the dust in snow could not be modelled, inducing a discrepancy between the modelled and the measured spectrum in the visible wavelengths. The referee is right to say the discrepancy between the modelled and measured spectrum can be attributed to a large number of approximations and assumptions in the model, but one of the assumption that has the most significant impact is to use only one type of impurity. In future study, it should be possible to estimate two effective impurities content one for "black" impurities and one

for "red" impurities but this is beyond the scope of the present study. The authors also agree that the whole paragraph was unclear. Consequently, it was rewritten as follows:

"
(…) estimated using Eq. 1**. Illustration of Step 2 is provided for three spectra in Fig. 2b.**
After these steps, spectra are filtered based on the root mean square deviation (RMSD) between **the measured spectrum and the optimal spectrum calculated with Eq. 8** with a threshold of 0.022. **This filtering ensures that the measurement artefacts are reasonably accounted for in the optimal snow surface properties estimation. The threshold value is higher than the one used in Picard et al., 2016a and was set to account for the discrepancies between the modelled and measured albedo spectra due among others to the impurity type assumption in the model. Indeed, the model used in this study only includes black carbon. Alpine snowpacks are frequently affected by deposition of Saharan dust (e.g. Di Mauro et al., 2015) that has a reddish spectral signature. The presence of red dust in snow can induce discrepancies between the modelled spectrum with black carbon only and the measured spectrum (fig. 4 in Warren and Wiscombe, 1982). This "dusty" pattern is clearly visible on the black measured spectrum of Fig. 2b in the visible wavelengths (400-500 nm).** "

**8** - p.8, lines 4-6: I recommend the authors to explain the motivation of Step 3, which looks unnecessary if they know in advance the slope and aspect of the surface. Also, I would explain here why the slope can change (with precipitation and snowdrift).

Thanks for this comment, we agree that the text was too short here.
As explained p8, line 5, step 3 is not strictly necessary but since the slope and aspect evolve during the season and since we have no direct measurements of these parameters, this step enables us to indirectly validate the physical consistency of K estimates.

The text has been consequently modified as follows:
"*Step 3 : Estimate daily optimal surface slope and aspect*
**Section 3.3 explains how the measured albedo varies with the surface slope and aspect. The slope and aspect below the sensor evolves in time with respect to precipitation, melt and snow transportation by the wind. Unfortunately no measurement of surface slope and aspect is available during this winter season. Consequently**, using K diurnal cycles and Equation 7, we estimate daily optimal values of surface slope angle and aspect for fully clear sky days. **This step is not required for the estimation of optimal SSA and Cimp** but it indirectly validates that K optimization has not compensated for other artefacts than slope. **In other words, estimating physically consistent values of daily slope and aspect from K diurnal cycles further ensures the consistency of the spectra correction."**

**9** - p.8, lines 9-10: How is the filtering of the spectra done? Do the authors remove the spectra that show a specific features in the moving window average?

Sorry the sentence was misleading. There is no filtering of the spectrum expect on the RMSD value as explained in Figure 11 legend.

The sentence has been modified as follows:
"The spectra are first **averaged** using a 20 nm moving window in order to reduce noise before minimum calculation."

**10** - p.8, lines 18: ": : :only slightly lower than the ideal value of one". This is not a correct expression,

as in my understanding A=1 is not an "ideal value" of A, but rather it indicates that the instrument had an ideal response. So, the concept would better be expressed by stating that a value of A close to 1 indicates that the measurement apparatus has caused only a small deviation from the ideal response of the instrument.

Agree, the sentence has been changed as follows:

"The distribution has a median value of 0.943. **This value is close to one, which indicates that the actual instrumental response deviates only slightly from the ideal one.**"

**11** - p.8, line 20: please remove "presence of"
Agree, the sentence has been changed accordingly.

**12** - p.8, lines 25-26: ": : : diurnal cycle with higher values in the morning and lower values in the afternoon". From Fig 4a I see that K is higher in the afternoon! Please consider redrawing Fig 4a, with just 24 hours in the x-axis to show better the diurnal cycle, and maybe the time series of K in just two selected days, one with good agreement and one will poor agreement.

p.8, lines 25-26 has been modified accordingly. Thanks for noticing the typo.
Fig 4a has also been modified as requested.
We also consequently modified the text p8 lines 25-26 :
"It also illustrates the good agreement between the optimal K and simulated K (blue crosses, step 3) **for March 11th (perfectly clear day) and the poorest agreement for March 10th where clouds were detected especially in the afternoon**. "

**13** - In Fig. 4b, please replace the symbol "â°U ¸ e" with "(degrees)" in the y-labels for slope and aspect.

Agree, Fig. 4b has been changed accordingly.

**14** - p. 9, line 2: "The seasonal evolution of slope and aspect seems to be related to snow evolution". It has to be, what else could cause a change in slope and aspect? So, I would replace "seems to be" with "is".
Agree, the sentence has been changed accordingly.

**15** - p. 9, line 21: "have been done under" should perhaps be "have been obtained under".
Agree, the sentence has been changed accordingly.

**16** - p. 11, lines 14-16: The text is too difficult to follow. The authors need to provide some interpretation, and also explain the applied relationship between SWE and snow density.

Agree the text has been modified as follows:
'It shows that the higher the SSA and Cimp, the higher the contribution of the uppermost centimeters **of the snowpack to the albedo value. For instance for low SSA (5 m$^2$ kg$^{-1}$) and high Cimp (500 ng g$^{-1}$), the top 10 kg m$^{-2}$ of the snowpack contributes to more than 80% of the signal. For a density of 400 kg m$^{-3}$, this corresponds to the uppermost 5 cm. For higher SSA (40 m$^2$ kg$^{-1}$) and lower Cimp (10 ng g$^{-1}$), the top 30 kg m$^{-2}$ of the snowpack contributes to more than 80% of the signal. For a density of 200 kg m$^{-3}$ , this corresponds to the uppermost 15 cm. This can be explained since (i) the higher the SSA the lower the light penetration depth in the snowpack and (ii) the presence of impurities shortens the light penetration depth in the snowpack (e.g. Libois et al., 2013)."**

**17** - p. 12, line 1: "2014-04-10" should perhaps be "2014-04-02"?
Agree, the date has been changed accordingly.

**18** - p.12, line 8-9: "The grey area corresponds to the 15% uncertainty estimated in Picard et al. (2016a)". Please clarify what this uncertainty refers to. In Fig 9c it appears as a constant value throughout the observational period rather than a percentage of SSA (which varied in the range _5-60 m2/kg).

We were referring to the estimated uncertainty on the optimal SSA value provided in Picard et al., 2016.
The sentence has been modified as follows :
"The grey area corresponds to the 15% uncertainty estimated in Picard et al. (2016a) **for the optimal SSA retrieval in Antarctica**."
Note that the y-axis in Fig. 9c is already expressed in % of SSA that's why we keep a constant value.

**19** - p.12, lines 24-29: I did not understand this paragraph. The authors write "If the RMSD is larger than the RMSD over the whole spectrum, Cimp is represented in red". What is the relevance of this? And what is the reasoning behind the application of this criterion? The message of this paragraph is totally unclear to me.

The authors agree that this was insufficiently explained. See also modifications done in response to comment 7 that provides additional details on that.
The text lines 24-29 was consequently modified as follows:
'If **this RMSD in the visible wavelengths (400-500 nm)** is larger than the RMSD over the whole spectrum, cimp is represented in red. **This indeed indicates that the predicted and modelled spectrum agrees well except in the 400-500 nm wavelengths as illustrated by the black spectrum in Fig. 2b and that the prevailing impurities at that date may have a reddish spectral signature.**'

**20** - Fig 9: In panel (a), I think that morning and afternoon symbols are inverted.
Thanks for noticing this. The figure legend and caption have been modified.

---

## Author Comment (AC3) · 10 Feb 2017

C. He

cenlinhe@atmos.ucla.edu

Authors responses are enlighten in blue. Proposed changes in the manuscript are reported in bold.

The authors measured snow spectral albedo in the visible/NIR range at an alpine site and highlighted the effects of snow specific surface area, impurity content, presence of liquid water, and slope on variations of spectral snow albedo. I have a short comment. In addition to the factors mentioned by the authors, recent studies also showed that snow grain shape and how impurities mixed with snow grains are two critical factors in determining snow albedo (e.g., Liou et al., 2014; He et al., 2014). I suggest including these references and adding some discussions on this aspect, which would be very interesting.

References:

Liou, K. N., Takano, Y., He, C., Yang, P., Leung, L. R., Gu, Y., and Lee, W. L.: Stochastic parameterization for light absorption by internally mixed BC/dust in snow grains for application to climate models, J. Geophys. Res.-Atmos., 119, 7616-7632, doi:10.1002/2014jd021665, 2014.

He, C., Li, Q. B., Liou, K. N., Takano, Y., Gu, Y., Qi, L., Mao, Y. H., and Leung, L. R.: Black carbon radiative forcing over the Tibetan Plateau, Geophys. Res. Lett., 41, 7806-7813, doi:10.1002/2014gl062191, 2014.

The authors are grateful for this interesting comment. The authors agree that grain shape and location of impurities with respect to the ice matrix are critical factors in determining snow albedo. In this study, B shape factor was set to constant in agreement with results shown in Libois et al., 2014 and impurities are assumed to be outside of the ice matrix.
These assumptions are discussed in the introduction of the manuscript (page 2, line 11 for grain shape and page 2, line 18 for the location of impurities). We have now added the two above-mentioned references in the manuscript introduction.
A sentence on these limitations has also been added in the last section of the manuscript (page 14, line 3).
"The influence of surface roughness, incident and reflected radiations **and of the location of the impurities with respect to the ice matrix** deserves future work. "

---

## Author Response (AR2)

First of all the authors would like to thank the two reviewers for their re-review of the manuscript.

Authors comments are in blue (changes in the manuscript in bold blue) and reviewers comments in black.

**Report #2**

In the revised manuscript, the authors have significantly improved the clarity of the text by adding the requested details and explanations. All the comments I made in my first review were properly taken into account, and the requested corrections were made.

Thanks !

Now I only have a couple of minor corrections to suggest:
- p.6, line 8: "in absence" should perhaps be "due to the absence". The meaning of the sentence is quite different in the two cases…

Ok the sentence now reads : 'The accuracy of the simulated spectral to diffuse solar irradiance ratio has not been evaluated **due to the** absence of measurements'.

- p.7, line 14-15: "...with respect to what would happen for an horizontal surface, then the slope of the surface...". I pointed to the unclarity of this sentence also in my first review, and the authors have indeed provide some text that makes now possible to understand it, but still the sentence is tortuous. I would modify it as "...with respect to the solid angles that would apply in the case of an horizontal surface. With these assumptions, the slope of the surface...

Ok the sentence has been modified according to the reviewer comments and is now :

'In the following, we assume that (i) both diffuse solar radiation and reflected radiation are isotropic and (ii) the surface slope is small and local enough not to modify significantly the solid angles under which the incoming and reflected radiations are measured with respect **to the solid angles that would apply in the case** of an horizontal surface. **With these assumptions,** the slope of the surface only affects the effective sun zenith and azimuth angles and thus the direct solar irradiance (see details in App. A and B)'